# Rectifying Open-Set Object Detection: Proper Evaluation and a Taxonomy

## Abstract

Open-set object detection (OSOD), a task involving the detection of unknown objects while accurately detecting known objects, has recently gained attention. However, we identify a fundamental issue with the problem formulation employed in current OSOD studies. Inherent to object detection is knowing "what to detect," which contradicts the idea of identifying "unknown" objects. This sets OSOD apart from open-set recognition (OSR). This contradiction complicates a proper evaluation of methods' performance, a fact that previous studies have overlooked. Next, we propose a novel formulation wherein detectors are required to detect both known and unknown classes within specified super-classes of object classes. This new formulation is free from the aforementioned issues and has practical applications. Finally, we design benchmark tests utilizing existing datasets and report the experimental evaluation of existing OSOD methods. As a byproduct, we introduce a taxonomy of OSOD, resolving confusion prevalent in the literature. We anticipate that our study will encourage the research community to reconsider OSOD and facilitate progress in the right direction.

## 1 Introduction

Open-set object detection (OSOD) is the problem of correctly detecting known objects in images while adequately dealing with unknown objects (e.g., detecting them as unknown). Here, known objects are the class of objects that detectors have seen at training time, and unknown objects are those they have not seen before. It has attracted much attention recently [22, 5, 16, 14, 30, 15, 40].

Early studies [22, 21, 5] consider how accurately detectors can detect known objects, without being distracted by unknown objects present in input images, which we will refer to as OSOD-I in what follows. Recent studies [16, 14, 30, 15, 40] have shifted the focus to detecting unknown objects as well. They follow the studies of open-set recognition (OSR) [27, 2, 24, 33, 41] and aim to detect any arbitrary unknown objects while preserving detection accuracy for known-classes, which we will refer to as OSOD-II.

In this paper, we point out a fundamental issue with the problem formulation of OSOD, which many recent studies rely on, specifically OSOD-II as defined above. OSOD-II requires detectors to detect both known-class and unknown-class objects. However, since unknown-class objects belong to an open set and can encompass any arbitrary classes, it is impossible for detectors to be fully aware of what to detect and what not to detect during inference. To address this, a potential approach is to design a detector that detects any "objects" appearing in images and classifies them as either known

Table 1: Proposed categorization of OSOD problems. "Det. target" indicates the target of detection. K and U indicate known and unknown objects, respectively.

| Type | Det. target | Unknown | Evaluation |
|---|---|---|---|
| OSOD-I[22, 5] | K | Any classes | Feasible |
| OSOD-II[16, 15] | K+U | Any classes | Hard |
| **OSOD-III** | K+U | Any sub-classes in a known super-class | Feasible |

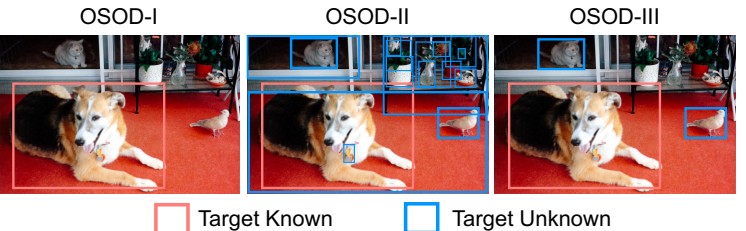

Target Known  Target Unknown

Figure 1: Illustration of OSOD-I, -II, and -III. OSOD-I: The interest is in detecting known objects without being distracted by unknown objects. OSOD-II: The interest is in detecting known and unknown objects as such. OSOD-III: The interest is in detecting known and unknown objects belonging to the same super-class as such.

or unknown classes. However, this approach is not feasible due to the ambiguity in the definition of "objects." For instance, should the tires of a car be considered as objects? It is important to note that such a difficulty does not arise in OSR since it is classification. Additionally, the aforementioned issue makes it hard to evaluate the performance of methods. Existing studies employ metrics such as A-OSE [22] and WI [5], which primarily measure the accuracy of *known* object detection (i.e., OSOD-I) and are not suitable for evaluating unknown object detection with OSOD-II.

Based on the above considerations, we propose a more practical formulation of OSOD, which we name OSOD-III. OSOD-III considers only unknown classes that belong to the same super-classes as the known classes, which distinguishes it from OSOD-II. This difference addresses the above issues of OSOD-II. Importantly, any method designed for OSOD-II can be applied to OSOD-III without modification. Figure 1 and Table 1 explain the concept of OSOD-III.

We design benchmark tests for OSOD-III using three existing datasets: Open Images [18], Caltech-UCSD Birds-200-2011 (CUB200) [34], and Mapillary Traffic Sign Dataset (MTSD) [7]. Thus, we evaluate the performance of four recent methods (designed for OSOD-II), namely ORE [16], Dropout Sampling (DS) [22], VOS [6], and OpenDet [15]. We also test a naive baseline method that classifies predicted boxes as known or unknown based on a simple uncertainty measure computed from predicted class scores. The results yield valuable insights. Firstly, the previous methods known for their good performance in metrics such as A-OSE and WI performed similarly or even worse than our simple baseline when they are evaluated with average precision (AP) in unknown object detection, a more appropriate performance metric. It is worth mentioning that our baseline employs standard detectors trained conventionally, without any additional training steps or extra architectures. Secondly, and more importantly, additional improvements are necessary to enable practical applications of OSOD(-III).

Our contributions are summarized as follows:

- We highlight a fundamental issue with the problem formulation used in current OSOD studies, which renders it ill-posed and makes proper performance evaluation difficult.

- In response, we introduce a new formulation of OSOD named OSOD-III, which addresses these concerns and offers practical applications.

Table 2: The class split employed in the standard benchmark test employed in recent studies of OSOD [16, 14, 15, 39, 40, 35]. Split1 consists of 20 PASCAL VOC classes. Split2, 3, and 4 consist of all the COCO classes but those of Split1. A typical setting is to use Split1 as known categories and Split2-4 as unknown categories. Note the dissimilarity between the known and unknown categories.

| | Split1 | Split2 | Split3 | Split4 |
|---|---|---|---|---|
| Classes | PASCAL VOC objects (20) | Outdoor(5), Accessories(5), Appliance(5), Animal(4), *Truck* | Sports(10), Food(10) | Electronic(5), Indoor(7), Kitchen(6), Furniture(2) |

- We develop benchmark tests using existing public datasets. Our experimental evaluation of existing OSOD methods demonstrates their unsatisfactory performance levels.

## 2 Rethinking Open-set Object Detection

### 2.1 Formalizing Problems

We first formulate the problem of open-set object detection (OSOD). Previous studies refer to two different problems as OSOD without clarification. We use the names of OSOD-I and -II to distinguish the two, which are defined as follows.

**OSOD-I** *The goal is to detect all instances of known objects in an image without being distracted by unknown objects present in the image. We want to avoid mistakenly detecting unknown object instances as known objects.*

**OSOD-II** *The goal is to detect all instances of known and unknown objects in an image, identifying them correctly (i.e., classifying them to known classes if known and to the "unknown" class otherwise).*

OSOD-I and -II both consider applying a closed-set object detector (i.e., a detector trained on a closed-set of object classes) to an open-set environment where the detector encounters objects of unknown class. Their difference is whether or not the detector detects unknown objects. OSOD-I does not; its concern is with the accuracy of detecting known objects. This problem is first studied in [5, 21, 22]. On the other hand, OSOD-II detector detects unknown objects as well, and thus their detection accuracy matters. OSOD-II is often considered as a part of open-world object detection (OWOD) [16, 14, 39, 30, 35].

The existing studies of OSOD-II rely on OWOD [16] for the problem formulation, which aims to generalize the concept of OSR (open-set recognition) to object detection. In OSR, *unknown* means "anything but known". Its direct translation to object detection is that *any arbitrary classes of objects but known objects can be considered unknown.* This formulation is reflected in the experimental settings employed in these studies. Table 2 shows the setting, which treats the 20 object classes of PASCAL VOC [8] as known classes and non-overlapping 60 classes from 80 of COCO [19] as unknown classes. This class split indicates the basic assumption that there is little relation between known and unknown objects.

However, this OSOD-II's formulation has an issue, making it ill-posed. It is because the task is detection. Detectors are requested to detect only objects that should be detected. It is a primary problem of object detection to judge whether or not something should be detected. What should not be detected include objects belonging to the background and irrelevant classes. Detectors learn to make this judgment, which is feasible for a closed set of object classes; what to detect is specified. However, this does not apply to OSOD-II, which aims at detecting also unknown objects defined as above. It is infeasible to specify what to detect and what not for any arbitrary objects in advance.

A naive solution to this difficulty is to detect *any* objects as long as they are "objects." However, it is not practical since defining what an object is itself hard. Figure 2 provides examples from COCO images. COCO covers only 80 object classes (shown in red rectangles in the images), and many unannotated objects are in the images (shown in blue rectangles). Is it necessary to consider every

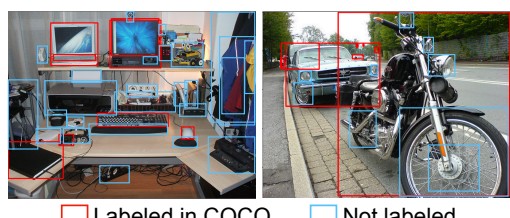

☐ Labeled in COCO  ☐ Not labeled

Figure 2: Example images showing that "object" is an ambiguous concept. It is impractical to cover an unlimited range of object instances with a finite set of predefined categories.

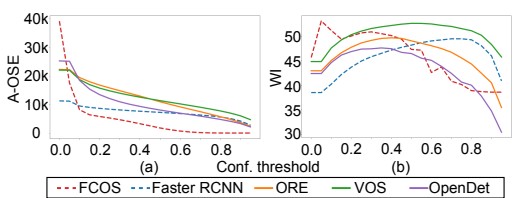

Figure 3: A-OSE (a) and WI (b) of different methods at different detector operating points. Smaller values mean better performance for both metrics. The horizontal axis indicates the confidence threshold for selecting bounding box candidates. Methods' ranking varies on the choice of the threshold.

one of them? Moreover, it is sometimes subjective to determine what constitutes individual "objects." For instance, a car consists of multiple parts, such as wheels, side mirrors, and headlights, which we may want to treat as "objects" depending on applications. This difficulty is well recognized in the prior studies of open-world detection [16, 14] and zero-shot detection [1, 20].

## 2.2 Metrics for Measuring OSOD Performance

The above difficulty also leads to make it hard to evaluate how well detectors detect unknown objects. The previous studies of OSOD employ two metrics for evaluating methods' performance, i.e., absolute open-set error (A-OSE) [22] and wilderness impact (WI) [5]. A-OSE is the number of predicted boxes that are in reality unknown objects but wrongly classified as known classes [22]. WI measures the ratio of the number of erroneous detections of unknowns as knowns (i.e., A-OSE) to the total number of detections of known instances, given by

$$\text{WI} = \frac{P_K}{P_{K \cup U}} - 1 = \frac{\text{A-OSE}}{\text{TP}_{known} + \text{FP}_{known}}, \tag{1}$$

where $P_K$ indicates the precision measured in the close-set setting; $P_{K \cup U}$ is that measured in the open-set setting; and $\text{TP}_{known}$ and $\text{FP}_{known}$ are the number of true positives and false positives for known classes, respectively.

These two metrics are originally designed for OSOD-I; they evaluate detectors' performance in open-set environments. Precisely, they measure how frequently a detector wrongly detects and misclassifies unknown objects as known classes (lower is better).

Nevertheless, previous studies of OSOD-II have employed A-OSE and WI as primary performance metrics. We point out that these metrics are pretty insufficient to evaluate OSOD-II detectors since they cannot evaluate the accuracy of detecting unknown objects, as mentioned above. They evaluate only one type of error, i.e., detecting unknown as known, and ignore the other type of error, detecting known as unknown.

In addition, we point out that A-OSE and WI are not flawless even as OSOD-I performance metrics. That is, they merely measure the detectors' performance at a single operating point; they cannot take the precision-recall tradeoff into account, the fundamental nature of detection. Specifically, previous studies [16] report A-OSE values for bounding boxes with confidence score $\geq 0.05$[1]. As for WI, previous studies [16, 14, 15, 30] choose the operating point of recall = 0.8. Thus, they show performance only partially since the setting is left to end users. Figures 3(a) and (b) show the profiles of A-OSE and WI, respectively, over the possible operating points of several existing OSOD-II detectors. It is seen that the ranking of the methods varies depending on the choice of confidence threshold.

---

[1]This is not clearly stated in the literature but can be confirmed with the public source code in GitHub repositories, e.g., https://github.com/JosephKJ/OWOD.

In summary, A-OSE and WI are insufficient for evaluating OSOD-II performance since i) they merely measure OSOD-I performance, i.e., only one of the two error types, and ii) they are metrics at a single operating point. To precisely measure OSOD-II performance, we must use average precision (AP), the standard metric for object detection, also to evaluate unknown object detection. It should be noted that while all the previous studies of OSOD-II report APs for known object detection, only a few report APs for unknown detection, such as [15, 35], probably because of the mentioned difficulty of specifying what unknown objects to detect and what not.

# 3 A More Practical Formulation

This section introduces another application formulation of OSOD. Although it has been overlooked in previous studies, we frequently encounter the scenario in practice. It is free from the fundamental issue of OSOD-II, enabling practical evaluation of methods' performance and probably making the problem easier to solve.

## 3.1 OSOD-III: Open at Class Level and Closed at Super-class level

Consider building a smartphone app that detects and classifies animal species. It is unrealistic to deal with all animal species at its initial deployment since there are too many classes. Thus, consider a strategy to start the app's service with a limited number of animal species; after its deployment, we want to add new classes by detecting unseen animal classes. To do this, we must design the detector to detect unseen animals accurately while correctly detecting known animals. After detecting unseen animals, we may collect their training data and retrain the detector using them. There will be many similar cases in real-world applications.

This problem is similar to OSOD-II; we want to detect unknown, novel animals. However, unlike OSOD-II, it is unnecessary to consider arbitrary objects as detection targets. In brief, we consider only animal classes; our detector does not need to detect any non-animal object, even if it has been unseen. In other words, we consider the set of object classes closed at the super-class level (i.e., animals) and open at the individual class level under the super-class.

We call this problem OSOD-III. The differences between OSOD-I, -II, and -III are shown in Fig. 1 and Table 1. The problem is formally stated as follows:

**OSOD-III** *Assume we are given a closed set of object classes belonging to a single super-class. Then, we want to detect and classify objects of these known classes correctly and to detect every unknown class object belonging to the same super-class and classify it as "unknown."*

It is noted that there may be multiple super-classes instead of a single. In that case, we need only consider the union of the super-classes. For the sake of simplicity, we only consider the case of a single super-class in what follows.

## 3.2 Properties of OSOD-III

While the applicability of OSOD-III is narrower than OSOD-II by definition, OSOD-III has two good properties[2].

One is that OSOD-III is free from the fundamental difficulty of OSOD-II, the dilemma of determining what unknown objects to detect and what to not. Indeed, the judgment is clear with OSOD-III; unknowns belonging to the known super-class should be detected, and all other unknowns should not. As a result, OSOD-III no longer suffers from the evaluation difficulty. The clear identification of detection targets enables the computation of AP also for unknown objects.

---

[2]Any OSOD-III problems can be interpreted as OSOD-II. However, it should always be beneficial to formulate it as OSOD-III if possible.

The other is that detecting unknowns will arguably be easier owing to the similarity between known and unknown classes. In OSOD-II, unknown objects can be arbitrarily dissimilar from known objects. In OSOD-III, known and unknown objects share their super-class, leading to their visual similarity. It should be noted here that what we regard as a super-class is arbitrary; there is no mathematical definition. However, as far as we consider reasonable class hierarchy as in WordNet/ImageNet [9, 4] , we may say that the sub-classes will share visual similarities.

# 4 Experimental Results

Based on the above formulation, we evaluate the performance of existing OSOD methods on the proposed OSOD-III scenario. In the following section, we first introduce our experimental settings to simulate the OSOD-III scenario and then report the evaluation results.

## 4.1 Experimental Settings

### 4.1.1 Datasets

We use the following three datasets for the experiments: Open Images Dataset v6 [18], Caltech-UCSD Birds-200-2011 (CUB200) [34], and Mapillary Traffic Sign Dataset (MTSD) [7]. For each, we split classes into known/unknown and images into training/validation/testing subsets as explained below. Note that one of the compared methods, ORE [16], needs validation images (i.e., example unknown-class instances), which may be regarded as leakage in OSOD problems. This does not apply to the other methods.

**Open Images** Open Images [18] contains 1.9M images of 601 classes of diverse objects with 15.9M bounding box annotations. It also provides the hierarchy of object classes in a tree structure, where each node represents a super-class, and each leaf represents an individual object category. For instance, a leaf *Polar Bear* has a parent node *Carnivore*. We choose two super-classes, *Animal* and *Vehicle*, in our experiments because of their appropriate numbers of sub-classes, i.e., 96 and 24 in the "Animal" and "Vehicle" super-class, respectively. We split these sub-classes into known and unknown classes. To mitigate statistical biases, we consider four random splits and select one for a known-class set and the union of the other three for an unknown-class set.

We construct the training/validation/testing splits of images based on the original splits provided by the dataset. Specifically, we choose the images containing at least one known-class instance from the original training and validation splits. We choose the images containing either at least one known-class instance or at least one unknown-class instance from the original testing split. For the training images, we keep annotations for the known objects and eliminate all other annotations including unknown objects. It should be noted that there is a risk that those removed objects could be treated as the "background" class. For the validation and testing images, we keep the annotations for known and unknown objects and remove all other irrelevant objects. See the supplementary material for more details.

**CUB200** Caltech-UCSD Birds-200-2011 (CUB200) [34] is a 200 fine-grained bird species dataset. It contains 12K images, for each of which a single box is provided. We split the 200 classes randomly into four splits, each with 50 classes. We then choose three to form a known-class set and treat the rest as an unknown-class set. We construct the training/validation/testing splits similarly to Open Images with two notable exceptions. One is that we create the training/validation/test splits from the dataset's original training/validation splits. This is because the dataset does not provide annotation for the original test split. The other is that we remove all the images containing unknown objects from the training splits. This will make the setting more rigorous. See the supplementary material for more details.

**MTSD** Mapillary Traffic Sign Dataset (MTSD) [7] is a dataset of 400 diverse traffic signs from different regions around the world. It contains 52K street-level images with 260K manually annotated

Table 3: Detection accuracy of known ($\mathrm{AP}_{known}$) and unknown objects ($\mathrm{AP}_{unk}$) of different methods on four benchmark tests (i.e., OpenImages-Animal/Vehicle, CUB200, and MTSD), for each of which the averages over all the splits are shown; see the supplementary material for more details.

| | Datasets | | | | | | | |
| --- | --- | --- | --- | --- | --- | --- | --- | --- |
| | Open Images-Animal | | Open Images-Vehicle | | CUB200 | | MTSD | |
| | $\mathrm{AP}_{known}$ | $\mathrm{AP}_{unk}$ | $\mathrm{AP}_{known}$ | $\mathrm{AP}_{unk}$ | $\mathrm{AP}_{known}$ | $\mathrm{AP}_{unk}$ | $\mathrm{AP}_{known}$ | $\mathrm{AP}_{unk}$ |
| ORE [16] | $37.6 \pm 2.8$ | $15.6 \pm 2.7$ | $33.7 \pm 8.5$ | $0.3 \pm 0.1$ | $53.2 \pm 1.3$ | $19.8 \pm 2.2$ | $41.2$ | $0.4 \pm 0.3$ |
| DS [22] | $41.1 \pm 2.9$ | $15.0 \pm 2.5$ | $40.1 \pm 7.9$ | $2.7 \pm 2.3$ | $61.5 \pm 0.9$ | $21.5 \pm 1.1$ | $50.4$ | $5.1 \pm 1.7$ |
| VOS [6] | $39.5 \pm 2.2$ | $16.0 \pm 1.8$ | $40.9 \pm 7.8$ | $9.1 \pm 2.2$ | $59.4 \pm 1.0$ | $8.7 \pm 0.6$ | $49.1$ | $4.7 \pm 1.5$ |
| OpenDet [15] | $36.9 \pm 8.1$ | $33.0 \pm 4.5$ | $38.7 \pm 7.8$ | $14.4 \pm 3.3$ | $63.3 \pm 1.1$ | $27.0 \pm 3.0$ | $51.8$ | $9.9 \pm 3.9$ |
| FCOS [32] | $30.3 \pm 4.7$ | $41.8 \pm 3.6$ | $30.7 \pm 12.0$ | $18.7 \pm 4.5$ | $53.5 \pm 2.1$ | $24.7 \pm 1.3$ | $41.7$ | $4.4 \pm 1.6$ |
| Faster RCNN [26] | $37.8 \pm 3.1$ | $35.3 \pm 3.9$ | $39.9 \pm 8.7$ | $17.0 \pm 5.2$ | $62.2 \pm 1.0$ | $24.2 \pm 1.9$ | $50.0$ | $3.1 \pm 1.2$ |

traffic sign instances. For the split of known/unknown classes, we consider a practical use case of OSOD-III, where a detector trained using the data from a specific region is used in another region, which might have unknown traffic signs. As the dataset does not provide region information for each image, we divide the 400 traffic sign classes into clusters based on their co-occurrence in the same images. Specifically, we apply normalized graph cut [28] to obtain three clusters, ensuring any pairs of the clusters share the minimum co-occurrence. We then use the largest cluster as a known-class set (230 classes). Denoting the other two clusters by unknown1 (55) and unknown2 (115), we test three cases, i.e., using either unknown1, unknown2, or their union (unknown1+2) for an unknown-class set. We report the results for the three cases. We create the training/validation/testing splits in the same way as CUB200. See the supplementary material for more details.

### 4.1.2 Evaluation

As discussed earlier, the primary metric for evaluating object detection performance is average precision (AP) [10, 8]. Although we must use AP for unknown detection, the issue with OSOD-II makes it impractical. OSOD-III is free from the issue, and we can use AP for unknown object detection. Therefore, following the standard evaluation procedure of object detection, we report AP over IoU in the range $[0.50, 0.95]$ for known and unknown object detection.

### 4.2 Compared Methods

In our benchmark testing, we consider four state-of-the-art methods; ORE [16], Dropout Sampling (DS) [22], VOS [6], and OpenDet [15]. Although these methods were originally developed for OSOD-II, they can be applied to OSOD-III without modification. See the supplementary material for more details of each method's configurations.

In addition to these existing methods, we also consider a naive baseline for comparison. It merely uses the class scores that standard detectors predict for each bounding box. It relies on an expectation that unknown-class inputs should result in uncertain class prediction. Thus we look at the prediction uncertainty to judge if the input belongs to known/unknown classes. Specifically, we calculate the ratio of the top-1 and top-2 class scores for each candidate bounding box and compare it with a pre-defined threshold $\gamma$; we regard the input as unknown if it is smaller than $\gamma$ and as known otherwise. We use the sum of the top three class scores for the unknown object detection score. In our experiments, we employ two detectors, FCOS [32] and Faster RCNN [26]. We use ResNet50-FPN as their backbone, following the above methods. For Open Images, we set $\gamma = 4.0$ for FCOS and $\gamma = 15.0$ for Faster RCNN. For CUB200 and MTSD, we set $\gamma = 1.5$ for FCOS and $\gamma = 3.0$ for Faster RCNN. We need different thresholds due to the difference in the number of classes and the output layer design, i.e., logistic vs. softmax. We report the sensitivity to the choice of $\gamma$ in the supplementary material.

### 4.3 Results

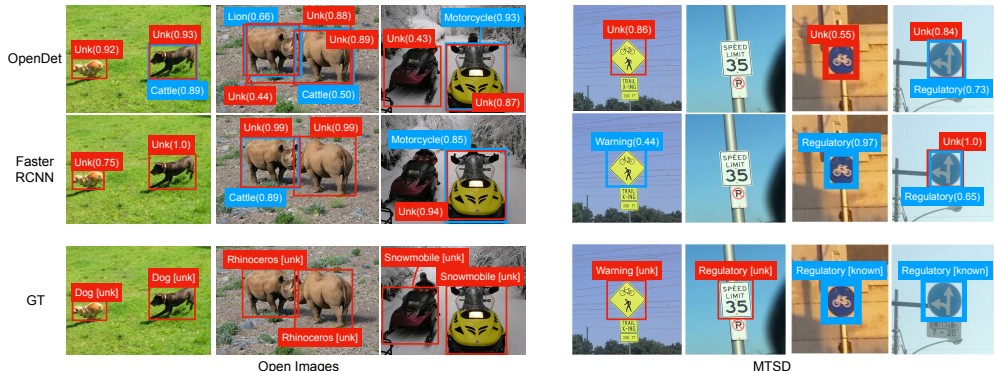

Figure 4: Example outputs of OpenDet [15] and our baseline method with Faster RCNN [26] for Open Images with Animal and Vehicle super-classes and MTSD, respectively. Red and blue boxes indicate detected unknown-class and known-class objects, respectively; "Unk" means "unknown".

Table 3 presents the results, specifically mAP for known-class objects ($AP_{known}$) and AP for unknown objects ($AP_{unk}$). The table shows the average and standard deviation values across all splits for each dataset. Performance results for individual splits can be found in the supplementary material.

For Open Images dataset, we can see from the table that the compared methods attain similar $AP_{known}$ (except the FCOS-based baseline due to the difference in the base detector). However, they show diverse performances in unknown object detection measured by $AP_{unk}$. Specifically, ORE, DS, and VOS yield inferior performance. While the rest of the methods achieve much better performance, we can observe that the two baseline methods outperform OpenDet, the current state-of-the-art. This good performance of these baseline methods is remarkable, considering that they do not require additional training or mechanism dedicated to unknown detection.

For the results of CUB200, we have similar observations with a few minor differences. Differences are that ORE works better for this dataset and OpenDet achieves the best performance. However, the gap between OpenDet and the baseline methods is not large with $AP_{known}$ and $AP_{unk}$.

Similar to the other datasets, for MTSD, all the methods maintain the good performance of known object detection; $AP_{known}$'s are high. OpenDet performs the best unknown detection performance with noticeable margins to others for this dataset.

Figure 4 shows selected examples of detection results by OpenDet and our baseline with Faster RCNN. There are erroneous detection results of treating unknown as known and vice versa, in addition to simple false negatives of unknown detection. These are consistent with the quantitative results in Table 3, indicating the unsatisfactory performance of existing methods.

## 4.4 Analysis on Failure Cases

The above results indicate that the detectors frequently misclassify between known and unknown instances. To address this, we examined the effect of applying non-maximum suppression (NMS) to these detectors.

In the above experiments, NMS was applied individually to each category, both known and unknown. This is consistent with the standard object detection procedure where NMS is typically used among the predicted bounding boxes (BBs) of a specific category. As shown in Fig 4, overlapping BBs between known and unknown categories remained. However, the appropriateness of treating the unknown category similarly to known categories in OSOD is debatable. As such, we expanded our approach to apply NMS across both known and unknown category predictions.

Figure 5 shows the mAP for the known category predictions and AP for the unknown, evaluated at varying IoU thresholds for NMS. In Fig 5, an IoU threshold of 1.0 represents results obtained

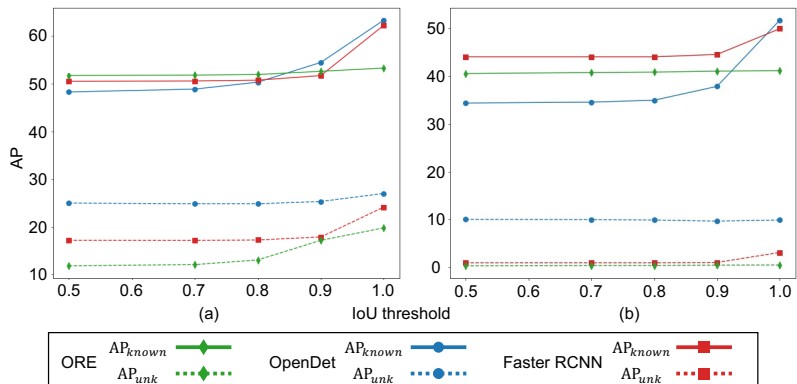

Figure 5: Detection accuracy at various IoU thresholds for NMS between known and unknown predictions: mAP for known categories and AP for unknown. The results for (a) CUB200 and (b) MTSD.

without NMS between known and unknown predictions. Results for other values reflect the impact of NMS. It is clear that aggressive NMS reduces APs for both categories. This observation suggests two things: i) Predicted known and unknown BBs frequently overlap, and ii) The scores of these bounding boxes do not consistently reflect prediction accuracy. Ideally, when BBs overlap, the highest scoring one should indicate the correct prediction. However, in our results, bounding boxes that misclassify known (or unknown) instances often score higher than the accurate ones. In summary, while the detectors are adept at detecting unknown instances, they regularly misidentify between known and unknown instances.

## 5   Related Work

### 5.1   Open-set Recognition

For the safe deployment of neural networks, open-set recognition (OSR) has attracted considerable attention. The task of OSR is to accurately classify known objects and simultaneously detect unseen objects as unknown. Scheirer *et al.* [27] first formulated the problem of OSR, and many following studies have been conducted so far [2, 12, 23, 31, 24, 29, 33, 41].

The work of Bendale and Boult [2] is the first to apply deep neural networks to OSR. They use outputs from the penultimate layer of a network to calibrate its prediction scores. Several studies [12, 23, 17] found generative models are effective for OSR, where unseen-class images are synthesized and used for training. Another line of OSR studies focuses on a reconstruction-based method using latent features [38, 36], class conditional auto-encoder [24], and conditional gaussian distributions [31].

### 5.2   Open-set Object Detection

We can categorize existing open-set object detection (OSOD) problems into two scenarios, OSOD-I and -II, according to their different interest in unknown objects, as we have discussed in this paper.

**OSOD-I**   Early studies treat OSOD as an extension of OSR problem [22, 21, 5]. They aim to correctly detect every known object instance and avoid misclassifying any unseen object instance into known classes. Miller *et al.* [22] first utilize multiple inference results through dropout layers [11] to estimate the uncertainty of the detector's prediction and use it to avoid erroneous detections under open-set conditions. Dhamija *et al.* [5] investigate how modern CNN detectors behave in an open-set environment and reveal that the detectors detect unseen objects as known objects with a high confidence score. For the evaluation, researchers have employed A-OSE [22] and WI [5] as the primary metrics to measure the accuracy of detecting known objects. They are designed to measure how frequently a detector wrongly detects and classifies unknown objects as known objects.

**OSOD-II**  More recent studies have moved in a more in-depth direction, where they aim to correctly detect/classify every object instance not only with the known class but also with the unknown class. This scenario is often considered a part of open-world object detection (OWOD) [16, 14, 39, 30, 35]. In this case, the detection of unknown objects matters since it considers updating the detectors by collecting unknown classes and using them for retraining. Joseph *et al*. [16] first introduces the concept of OWOD and establishes the benchmark test. Many subsequent works have strictly followed this benchmark and proposed methods for OSOD. OW-DETR [14] introduces a transformer-based detector (i.e., DETR [3, 42]) for OWOD and improves the performance. Han *et al*. [15] propose OpenDet and pay attention to the fact that unknown classes are distributed in low-density regions in the latent space. They then perform contrastive learning to encourage intra-class compactness and inter-class separation of known classes, leading to performance gain. Similarly, Du *et al*. [6] synthesize virtual unseen samples from the decision boundaries of gaussian distributions for each known class. Wu *et al*. [35] propose a further challenging task to distinguish unknown instances as multiple unknown classes.

### 5.3   Open Vocabulary Object Detection

It is noteworthy to highlight the difference/similarity between OSOD-III (which is formulated in this paper) and open vocabulary object detection (OVD) [37, 13, 25]. OVD involves detecting novel objects by providing only their names as texts (i.e., class names) without explicit training data (i.e., image-text pairs). Thus, one could argue that it shares some similarities with OSOD-III, as it requires detectors to detect novel objects within an assumed super-class. However, they are clearly different. Firstly, OVD provides information, albeit limited to texts, about the objects to detect, whereas OSOD-III provides no such information. Secondly, in OVD, the detector's backbone has the opportunity to learn about the novel objects during its pretraining phase, either explicitly (i.e., with direct image-text pairs) or implicitly (i.e., by aligning image and text feature spaces). In contrast, detectors for OSOD-III have no such opportunity to learn about the novel classes; what we assume for the super-class is not transferred to the detectors.

## 6   Conclusion and Discussions

In this paper, we have considered the problem of open-set object detection (OSOD). We categorize previous formulations of OSOD into two types: OSOD-I and OSOD-II. Firstly, we highlight the ill-posedness of OSOD-II, where it is difficult to determine what to detect and what not for unknown objects. This difficulty makes the evaluation infeasible; as a result, the previous studies employ insufficient metrics, A-OSE and WI, for evaluating methods' performance, designed originally for OSOD-I and not measuring the accuracy of unknown object detection.

We have then introduced a new scenario, OSOD-III. It considers the detection of unknown objects belonging to the same super-class as the known objects. This formulation is free from the above issues. We can determine what to detect or not in advance and then appropriately evaluate methods' performance using a standard AP metric for known and unknown detection. We have also designed benchmark tests tailored to the proposed scenario and evaluated the existing OSOD methods and a baseline method we designed in this paper on them. While they provide a few valuable insights, the main conclusion is that current methods attain only limited unknown detection performance. There is a lot of room for further improvement in OSOD-III.

The analysis in Sec. 4.4 indicates that future research should address the prevalent issue of misclassifying known and unknown instances. While detecting BBs of unknown instances isn't particularly challenging, the issue arises in classification: BBs predicted for unknown instances are often mislabeled as known, and vice-versa. Moreover, simply applying NMS to both known and unknown predictions isn't a comprehensive solution. The primary challenge appears to be in comparing their respective confidence scores. This discrepancy is likely because the scores aren't consistently calibrated between the known and unknown categories.

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
