# Supplementary material for "Rectifying Open-Set Object Detection: Proper Evaluation and a Taxonomy"

## A  Access to Our Dataset

Our datasets can be accessed at https://github.com/rsCPSyEu/OSOD-III.git.

## B  Details of the Datasets

We used three datasets in our experiments, i.e., Open Images dataset [10], Caltech-UCSD Birds-200-2011 (CUB200) [16], and Mapillary Traffic Sign Dataset (MTSD) [3]. Tables 4, 5, and 6 show their splits, based on which known/unknown classes are selected, and also those of training/validation/testing images. Tables 7, 8, and 9 provide lists of the classes for each split. Please check Sec 4.1.1 in the main paper as well.

Table 4: Details of the employed class splits for Open Images dataset. We treat one of the four as a known set and the union of the other three as an unknown set. Thus, there are four cases of known/unknown splits, for each of which we report the detection performance in Table 10.

|  | Animal | | | | Vehicle | | | |
|---|---|---|---|---|---|---|---|---|
|  | Split1 | Split2 | Split3 | Split4 | Split1 | Split2 | Split3 | Split4 |
| num of known categories | 24 | 24 | 24 | 24 | 6 | 6 | 6 | 6 |
| train images | 44,379 | 38,914 | 39,039 | 18,478 | 43,270 | 26,860 | 3,900 | 6,300 |
| validation images | 1,104 | 2,353 | 1,248 | 849 | 1,370 | 503 | 178 | 322 |
| test images | 15,609 | | | | 6,991 | | | |

Table 5: Details of the employed class splits for Caltech-UCSD Birds-200-2011 (CUB200) dataset. We treat the union of three of the four as known classes and the rest as unknown classes. Each split corresponds to the results shown in Table 11.

|  | Split1 | Split2 | Split3 | Split4 |
|---|---|---|---|---|
| num of unknown classes | 50 | 50 | 50 | 50 |
| train images | 4,109 | 4,116 | 4,120 | 4,120 |
| validation images | 500 | 500 | 500 | 500 |
| test images | 5,794 | | | |

Submitted to the 37th Conference on Neural Information Processing Systems (NeurIPS 2023) Track on Datasets and Benchmarks. Do not distribute.

Table 6: Details of the employed class splits for Mapillary Traffic Sign Dataset (MTSD). Each split corresponds to the results shown in Table 12.

|  | Unknown1 | Unknown2 | Unknown1+2 |
|---|---|---|---|
| num of classes | 55 | 115 | 170 |
| train images |  | 13, 157 |  |
| validation images |  | 1, 000 |  |
| test images |  | 3, 896 |  |

Table 7: Classes contained in the employed splits for Open Images [10] with the super-classes "Animal" (first column) and "Vehicle" (second column), respectively.

|  | Animal | Vehicle |
|---|---|---|
| Split1 (24/6) | Starfish / Deer / Tick / Lynx / Monkey / Squirrel / Koala / Fox / Spider / Scorpion / Rabbit / Hamster / Woodpecker / Snail / Brown bear / Polar bear / Lion / Bull / Shrimp / Panda / Chicken / Sparrow / Cattle / Lobster | Bicycle / Golf cart / Van / Taxi / Airplane / Motorcycle |
| Split2 (24/6) | Sea lion / Mule / Lizard / Raccoon / Butterfly / Hippopotamus / Kangaroo / Frog / Harbor seal / Red panda / Antelope / Ant / Sheep / Dog / Magpie / Teddy bear / Oyster / Otter / Seahorse / Caterpillar / Worm / Zebra / Jaguar (Animal) / Rays and skates | Train / Truck / Barge / Gondola / Rocket / Bus |
| Split3 (24/6) | Tortoise / Skunk / Blue jay / Rhinoceros / Turkey / Falcon / Dinosaur / Bat (Animal) / Squid / Giraffe / Owl / Armadillo / Swan / Duck / Goose / Camel / Horse / Tiger / Goldfish / Cat / Shark / Parrot / Leopard / Goat | Submarine / Jet ski / Unicycle / Snowmobile / Cart / Tank |
| Split4 (24/6) | Dragonfly / Ladybug / Raven / Penguin / Hedgehog / Mouse / Snake / Jellyfish / Porcupine / Ostrich / Elephant / Dolphin / Alpaca / Crab / Eagle / Isopod / Cheetah / Sea turtle / Whale / Bee / Canary / Pig / Crocodile / Centipede | Canoe / Helicopter / Wheelchair / Ambulance / Segway / Limousine |

Table 8: Classes contained in the employed splits for CUB200 [16].

| | |
|---|---|
| Split1 (50) | Black footed Albatross / Laysan Albatross / Least Auklet / Red winged Blackbird / Yellow headed Blackbird / Indigo Bunting / Spotted Catbird / Brandt Cormorant / Red faced Cormorant / Shiny Cowbird / Brown Creeper / Yellow billed Cuckoo / Purple Finch / Acadian Flycatcher / Scissor tailed Flycatcher / Vermilion Flycatcher / Western Grebe / Ivory Gull / Ruby throated Hummingbird / Rufous Hummingbird / Green Jay / Belted Kingfisher / Pied Kingfisher / Pacific Loon / Mallard / Western Meadowlark / Orchard Oriole / Scott Oriole / Whip poor Will / Loggerhead Shrike / Great Grey Shrike / Brewer Sparrow / Grasshopper Sparrow / Henslow Sparrow / Le Conte Sparrow / Cape Glossy Starling / Bank Swallow / Tree Swallow / Common Tern / Least Tern / Philadelphia Vireo / Wilson Warbler / Pileated Woodpecker / Red bellied Woodpecker / Red cockaded Woodpecker / Bewick Wren / Marsh Wren / Rock Wren / Winter Wren / Common Yellowthroat |
| Split2 (50) | Groove billed Ani / Crested Auklet / Parakeet Auklet / Bobolink / Lazuli Bunting / Gray Catbird / Fish Crow / Gray crowned Rosy Finch / Least Flycatcher / Gadwall / Blue Grosbeak / Heermann Gull / Ring billed Gull / Slaty backed Gull / Green Violetear / Pomarine Jaeger / Red breasted Merganser / Mockingbird / White breasted Nuthatch / Baltimore Oriole / Western Wood Pewee / American Pipit / Geococcyx / Baird Sparrow / House Sparrow / Field Sparrow / Seaside Sparrow / Vesper Sparrow / White throated Sparrow / Cliff Swallow / Scarlet Tanager / Summer Tanager / Elegant Tern / Forsters Tern / Green tailed Towhee / Brown Thrasher / Blue headed Vireo / White eyed Vireo / Bay breasted Warbler / Black and white Warbler / Golden winged Warbler / Nashville Warbler / Orange crowned Warbler / Palm Warbler / Pine Warbler / Swainson Warbler / Tennessee Warbler / Bohemian Waxwing / American Three toed Woodpecker / Carolina Wren |
| Split3 (50) | Sooty Albatross / Rhinoceros Auklet / Brewer Blackbird / Rusty Blackbird / Painted Bunting / Cardinal / Chuck will Widow / Pelagic Cormorant / Bronzed Cowbird / American Crow / Mangrove Cuckoo / Yellow bellied Flycatcher / Northern Fulmar / European Goldfinch / Boat tailed Grackle / Horned Grebe / Evening Grosbeak / Pigeon Guillemot / Herring Gull / Western Gull / Anna Hummingbird / Long tailed Jaeger / Gray Kingbird / Green Kingfisher / Horned Lark / Clark Nutcracker / Brown Pelican / Sayornis / Common Raven / White necked Raven / Black throated Sparrow / Chipping Sparrow / Clay colored Sparrow / Fox Sparrow / Savannah Sparrow / White crowned Sparrow / Barn Swallow / Black Tern / Caspian Tern / Sage Thrasher / Red eyed Vireo / Cape May Warbler / Chestnut sided Warbler / Kentucky Warbler / Mourning Warbler / Prairie Warbler / Yellow Warbler / Louisiana Waterthrush / Red headed Woodpecker / Cactus Wren |
| Split4 (50) | Yellow breasted Chat / Eastern Towhee / Black billed Cuckoo / Northern Flicker / Great Crested Flycatcher / Olive sided Flycatcher / Frigatebird / American Goldfinch / Eared Grebe / Pied billed Grebe / Pine Grosbeak / Rose breasted Grosbeak / California Gull / Glaucous winged Gull / Blue Jay / Florida Jay / Dark eyed Junco / Tropical Kingbird / Ringed Kingfisher / White breasted Kingfisher / Red legged Kittiwake / Hooded Merganser / Nighthawk / Hooded Oriole / Ovenbird / White Pelican / Horned Puffin / American Redstart / Harris Sparrow / Lincoln Sparrow / Nelson Sharp tailed Sparrow / Song Sparrow / Tree Sparrow / Artic Tern / Black capped Vireo / Warbling Vireo / Yellow throated Vireo / Black throated Blue Warbler / Blue winged Warbler / Canada Warbler / Cerulean Warbler / Hooded Warbler / Magnolia Warbler / Myrtle Warbler / Prothonotary Warbler / Worm eating Warbler / Northern Waterthrush / Cedar Waxwing / Downy Woodpecker / House Wren |

Table 9: Classes contained in the Unknown1 and Unkonwn2 splits of MTSD [3]. The rest are treated as known classes.

| | |
|---|---|
| Unknown1 (55) | complementary–chevron-left–g1 / complementary–chevron-right–g1 / complementary–maximum-speed-limit-20–g1 / complementary–maximum-speed-limit-25–g1 / complementary–maximum-speed-limit-30–g1 / complementary–maximum-speed-limit-35–g1 / complementary–maximum-speed-limit-40–g1 / complementary–maximum-speed-limit-45–g1 / complementary–maximum-speed-limit-50–g1 / complementary–maximum-speed-limit-55–g1 / complementary–maximum-speed-limit-70–g1 / complementary–maximum-speed-limit-75–g1 / information–highway-exit–g1 / information–safety-area–g2 / regulatory–detour-left–g1 / regulatory–keep-right–g6 / regulatory–no-overtaking–g5 / regulatory–weight-limit-with-trucks–g1 / warning–accidental-area-unsure–g2 / warning–bus-stop-ahead–g3 / warning–curve-left–g2 / warning–curve-right–g2 / warning–domestic-animals–g3 / warning–double-curve-first-left–g2 / warning–double-curve-first-right–g2 / warning–double-turn-first-right–g1 / warning–falling-rocks-or-debris-right–g2 / warning–falling-rocks-or-debris-right–g4 / warning–hairpin-curve-left–g1 / warning–hairpin-curve-right–g1 / warning–hairpin-curve-right–g4 / warning–horizontal-alignment-left–g1 / warning–horizontal-alignment-right–g1 / warning–horizontal-alignment-right–g3 / warning–junction-with-a-side-road-acute-right–g1 / warning–junction-with-a-side-road-perpendicular-left–g3 / warning–junction-with-a-side-road-perpendicular-right–g3 / warning–kangaloo-crossing–g1 / warning–loop-270-degree–g1 / warning–narrow-bridge–g1 / warning–offset-roads–g3 / warning–railroad-crossing-with-barriers–g2 / warning–railroad-intersection–g4 / warning–road-widens–g1 / warning–road-widens-right–g1 / warning–slippery-motorcycles–g1 / warning–slippery-road-surface–g2 / warning–steep-ascent–g7 / warning–trucks-crossing–g1 / warning–turn-left–g1 / warning–turn-right–g1 / warning–winding-road-first-left–g1 / warning–winding-road-first-right–g1 / warning–wombat-crossing–g1 / warning–y-roads–g1 / |
| Unknown2 (115) | complementary–both-directions–g1 / complementary–chevron-right–g3 / complementary–go-left–g1 / complementary–go-right–g1 / complementary–go-right–g2 / complementary–keep-left–g1 / complementary–keep-right–g1 / complementary–maximum-speed-limit-15–g1 / complementary–one-direction-left–g1 / complementary–one-direction-right–g1 / complementary–turn-left–g2 / complementary–turn-right–g2 / information–airport–g2 / information–bike-route–g1 / information–camp–g1 / information–gas-station–g1 / information–highway-interstate-route–g2 / information–hospital–g1 / information–interstate-route–g1 / information–lodging–g1 / information–parking–g3 / information–parking–g6 / information–trailer-camping–g1 / regulatory–bicycles-only–g2 / regulatory–bicycles-only–g3 / regulatory–do-not-block-intersection–g1 / regulatory–do-not-stop-on-tracks–g1 / regulatory–dual-lanes-go-straight-on-left–g1 / regulatory–dual-lanes-go-straight-on-right–g1 / regulatory–dual-lanes-turn-left-no-u-turn–g1 / regulatory–dual-lanes-turn-left-or-straight–g1 / regulatory–dual-lanes-turn-right-or-straight–g1 / regulatory–go-straight–g3 / regulatory–go-straight-or-turn-left–g2 / regulatory–go-straight-or-turn-left–g3 / regulatory–go-straight-or-turn-right–g3 / regulatory–keep-right–g4 / regulatory–lane-control–g1 / regulatory–left-turn-yield-on-green–g1 / regulatory–maximum-speed-limit-100–g3 / regulatory–maximum-speed-limit-25–g2 / regulatory–maximum-speed-limit-30–g3 / regulatory–maximum-speed-limit-35–g2 / regulatory–maximum-speed-limit-40–g3 / regulatory–maximum-speed-limit-40–g6 / regulatory–maximum-speed-limit-45–g3 / regulatory–maximum-speed-limit-50–g6 / regulatory–maximum-speed-limit-55–g2 / regulatory–maximum-speed-limit-65–g2 / regulatory–no-left-turn–g1 / regulatory–no-parking–g2 / regulatory–no-parking-or-no-stopping–g1 / regulatory–no-parking-or-no-stopping–g2 / regulatory–no-parking-or-no-stopping–g3 / regulatory–no-right-turn–g1 / regulatory–no-stopping–g2 / regulatory–no-stopping–g4 / regulatory–no-straight-through–g1 / regulatory–no-turn-on-red–g1 / regulatory–no-turn-on-red–g2 / regulatory–no-turn-on-red–g3 / regulatory–no-turns–g1 / regulatory–no-u-turn–g1 / regulatory–one-way-left–g2 / regulatory–one-way-left–g3 / regulatory–one-way-right–g2 / regulatory–one-way-right–g3 / regulatory–parking-restrictions–g2 / regulatory–pass-on-either-side–g2 / regulatory–passing-lane-ahead–g1 / regulatory–reversible-lanes–g2 / regulatory–road-closed–g2 / regulatory–roundabout–g2 / regulatory–stop–g1 / regulatory–stop-here-on-red-or-flashing-light–g1 / regulatory–stop-here-on-red-or-flashing-light–g2 / regulatory–text-four-lines–g1 / regulatory–triple-lanes-turn-left-center-lane–g1 / regulatory–truck-speed-limit-60–g1 / regulatory–turn-right–g3 / regulatory–turning-vehicles-yield-to-pedestrians–g1 / regulatory–wrong-way–g1 / warning–added-lane-right–g1 / warning–bicycles-crossing–g2 / warning–bicycles-crossing–g3 / warning–divided-highway-ends–g2 / warning–double-reverse-curve-right–g1 / warning–dual-lanes-right-turn-or-go-straight–g1 / warning–emergency-vehicles–g1 / warning–equestrians-crossing–g2 / warning–flaggers-in-road–g1 / warning–height-restriction–g2 / warning–junction-with-a-side-road-perpendicular-left–g4 / warning–pass-left-or-right–g2 / warning–pedestrians-crossing–g4 / warning–pedestrians-crossing–g9 / warning–playground–g1 / warning–playground–g3 / warning–railroad-crossing–g1 / warning–railroad-intersection–g3 / warning–road-narrows-left–g2 / warning–road-narrows-right–g2 / warning–roundabout–g25 / warning–school-zone–g2 / warning–shared-lane-motorcycles-bicycles–g1 / warning–stop-ahead–g9 / warning–texts–g1 / warning–texts–g2 / warning–texts–g3 / warning–traffic-merges-right–g1 / warning–traffic-signals–g3 / warning–trail-crossing–g2 / warning–two-way-traffic–g2 / |

# C   More Details of Experimental Settings

We provide a comprehensive description of the experimental configurations utilized in the evaluation of our main paper.

## C.1   Training

We train the models using the SGD optimizer with the batch size of 16 on 8 A100 GPUs. The number of epochs is 12, 80, and 60 for OpenImages, CUB200, and MTSD, respectively. We use the initial learning rate of $2.0 \times 10^{-2}$ with momentum $= 0.9$ and weight decay $= 1.0 \times 10^{-4}$. We drop a learning rate by a factor of 10 at $2/3$ and $11/12$ epoch. For Open Images and CUB200, we follow a common multi-scale training and resize the input images such that their shorter side is between 480 and 800, while the longer side is 1333 or less. At the inference time, we set the shorter side of input images to 800 and the longer side to less or equal to 1333. For MTSD, we apply similar scaling strategies to Open Images and CUB200 (i.e., multi-scale training and single-scale testing) but the scaling scheme; namely, the input size is doubled, e.g., the shorter side is between 960 and 1600 at training time. This aims to improve detection accuracy for the small-sized objects that frequently appear in MTSD.

We used the publicly available source code for the implementation of ORE[1] [9], Dropout Sampling (DS)[2] [12], VOS[3] [2], and OpenDet[4] [7]. We used mmdetection[5] [1] for FCOS [15] and detectron2[6] for Faster RCNN [14] to implement the baseline methods, respectively.

## C.2   Experimental Configurations for Compared Methods

As mentioned in Sec 4.2 of the main paper, our experiments involve four OSOD methods. Although these methods were originally developed for OSOD-II, they can be applied to OSOD-III without any modification. We provide a summary of their methods and present the corresponding configurations.

**ORE (Open World Object Detector)** [9] is initially designed for OWOD; it is capable not only of detecting unknown objects but also of incremental learning. We omit the latter capability and use the former as an open-set object detector. It employs an energy-based method to classify known/unknown; using the validation set, including unknown object annotations, it models the energy distributions for known and unknown objects. To compute AP for unknown objects, we use a detection score that ORE provides. Following the original paper [9], we employ Faster RCNN [14] with a ResNet50 backbone [8] for the base detector.

**DS (Dropout Sampling)** [12] uses the entropy of class scores to discriminate known and unknown categories. Specifically, during the inference phase, it employs a dropout layer [5] right before computing class logits and performs inference $n$ iterations. If the entropy of the average class logits over these iterations exceeds a threshold, the detected instance is assigned to the unknown category. The top-1 class score, calculated from the averaged class logits, is employed as the unknown score for computing unknown AP. Our base detector is Faster RCNN with ResNet50-FPN backbone [11]. Following the implementation of [7], we set the number of inference iterations $n$ to 30, the entropy threshold $\gamma_{ds}$ to 0.25, and the dropout layer parameter $p$ to 0.5, respectively.

**VOS (Virtual Outlier Synthesis)** [2] detects unknown objects by treating them as out-of-distribution (OOD) based on an energy-based method. Specifically, it estimates an energy value for each detected instance and judges whether it is known or unknown by comparing the energy with a threshold.

---

[1] https://github.com/JosephKJ/OWOD.git
[2] https://github.com/csuhan/opendet2.git
[3] https://github.com/deeplearning-wisc/vos.git
[4] https://github.com/csuhan/opendet2.git
[5] https://github.com/open-mmlab/mmdetection.git
[6] https://github.com/facebookresearch/detectron2

Table 10: Detection accuracy of known ($AP_{known}$) and unknown objects ($AP_{unk}$) of different methods for Open Images dataset, "Animal" and "Vehicle" super-classes. "Split-$n$" indicates that the classes of Split-$n$ are treated as known classes. "mean" is their average that is also shown in Table 3 in the main paper.

| | Animal | | | | | | | | | |
| --- | --- | --- | --- | --- | --- | --- | --- | --- | --- | --- |
| | Split1 | | Split2 | | Split3 | | Split4 | | mean | |
| | $AP_{known}$ | $AP_{unk}$ | $AP_{known}$ | $AP_{unk}$ | $AP_{known}$ | $AP_{unk}$ | $AP_{known}$ | $AP_{unk}$ | $AP_{known}$ | $AP_{unk}$ |
| ORE [9] | 40.4 | 17.4 | 34.8 | 13.0 | 40.4 | 19.1 | 34.8 | 13.0 | 37.6 ± 2.8 | 15.6 ± 2.7 |
| DS [12] | 44.0 | 19.0 | 36.8 | 12.3 | 43.3 | 14.0 | 40.2 | 14.6 | 41.1 ± 2.9 | 15.0 ± 2.5 |
| VOS [2] | 39.5 | 17.5 | 37.5 | 13.9 | 43.1 | 14.7 | 37.9 | 18.1 | 39.5 ± 2.2 | 16.0 ± 1.8 |
| OpenDet [7] | 42.4 | 34.9 | 23.2 | 25.8 | 43.0 | 37.9 | 39.0 | 33.5 | 36.9 ± 8.1 | 33.0 ± 4.5 |
| FCOS [15] | 35.0 | 44.4 | 30.8 | 35.6 | 32.6 | 43.7 | 22.6 | 43.6 | 30.3 ± 4.7 | 41.8 ± 3.6 |
| Faster RCNN [14] | 41.8 | 36.9 | 34.0 | 29.5 | 39.7 | 37.7 | 35.5 | 37.0 | 37.8 ± 3.1 | 35.3 ± 3.9 |
| | Vehicle | | | | | | | | | |
| | Split1 | | Split2 | | Split3 | | Split4 | | mean | |
| | $AP_{known}$ | $AP_{unk}$ | $AP_{known}$ | $AP_{unk}$ | $AP_{known}$ | $AP_{unk}$ | $AP_{known}$ | $AP_{unk}$ | $AP_{known}$ | $AP_{unk}$ |
| ORE [9] | 46.9 | 0.5 | 35.0 | 0.1 | 25.0 | 0.2 | 27.7 | 0.3 | 33.7 ± 8.5 | 0.3 ± 0.1 |
| DS [12] | 52.6 | 0.5 | 40.7 | 2.3 | 31.9 | 6.5 | 35.1 | 1.4 | 40.1 ± 7.9 | 2.7 ± 2.3 |
| VOS [2] | 53.2 | 7.4 | 41.9 | 7.1 | 32.8 | 9.4 | 35.7 | 12.6 | 40.9 ± 7.8 | 9.1 ± 2.2 |
| OpenDet [7] | 50.6 | 10.2 | 40.4 | 12.5 | 30.2 | 15.9 | 33.6 | 19.0 | 38.7 ± 7.8 | 14.4 ± 3.3 |
| FCOS [15] | 49.6 | 14.2 | 32.7 | 14.6 | 19.2 | 24.7 | 21.4 | 21.3 | 30.7 ± 12.0 | 18.7 ± 4.5 |
| Faster RCNN [14] | 51.0 | 10.5 | 42.0 | 15.2 | 31.0 | 22.1 | 35.7 | 20.2 | 39.9 ± 8.7 | 17.0 ± 5.2 |

Table 11: Detection accuracy for CUB200 [16]. See Table 10 for notations.

| | Split1 | | Split2 | | Split3 | | Split4 | | mean | |
| --- | --- | --- | --- | --- | --- | --- | --- | --- | --- | --- |
| | $AP_{known}$ | $AP_{unk}$ | $AP_{known}$ | $AP_{unk}$ | $AP_{known}$ | $AP_{unk}$ | $AP_{known}$ | $AP_{unk}$ | $AP_{known}$ | $AP_{unk}$ |
| ORE [9] | 51.3 | 18.1 | 53.6 | 21.8 | 54.4 | 17.7 | 53.6 | 21.6 | 53.2 ± 1.3 | 19.8 ± 2.2 |
| DS [12] | 61.7 | 19.6 | 61.2 | 22.2 | 62.8 | 22.2 | 60.4 | 21.8 | 61.5 ± 0.9 | 21.5 ± 1.1 |
| VOS [2] | 59.7 | 8.1 | 59.5 | 9.1 | 60.5 | 8.1 | 57.7 | 9.5 | 59.4 ± 1.0 | 8.7 ± 0.6 |
| OpenDet [7] | 63.9 | 23.1 | 63.6 | 30.0 | 63.9 | 26.3 | 61.6 | 28.6 | 63.3 ± 1.1 | 27.0 ± 3.0 |
| FCOS [15] | 55.0 | 23.0 | 55.2 | 26.1 | 50.6 | 25.0 | 53.0 | 24.6 | 53.5 ± 2.1 | 24.7 ± 1.3 |
| Faster RCNN [14] | 62.0 | 21.6 | 62.7 | 26.2 | 63.2 | 24.0 | 60.8 | 24.8 | 62.2 ± 1.0 | 24.2 ± 1.9 |

We use the energy value to compute unknown AP. We choose Faster RCNN with ResNet50-FPN backbone [11], following the paper.

**OpenDet (Open-set Detector)** [7] is the current state-of-the-art on the popular benchmark test designed using PASCAL VOC/COCO shown in Table 2, although the methods' performance is evaluated with inappropriate metrics of A-OSE and WI. OpenDet provides a detection score for unknown objects, which we utilize to compute AP. We use the authors' implementation, which employs Faster RCNN based on ResNet50-FPN for the base detector.

## D  Additional Experimental Results

### D.1  Detection Accuracy for Individual Splits

Tables 10, 11, and 12 show detection accuracy of known $AP_{known}$ and unknown $AP_{unk}$ for each split and their averages. The classes denoted as "Split-$n$" and "Unknown-$n$" in the results correspond to the class sets specified in Tables 7, 8, and 9.

### D.2  Results of H-score

To facilitate easier comparisons of detection accuracy, we report H-score [4] as a comprehensive evaluation metric. H-score was originally designed in open-set recognition (OSR) task as a harmonic mean of known and unknown categories. We adopt this metric to object detection, calculating a harmonic mean of average precision (AP) for these two distinct categories. Tables 13, 14, and 15 show the results for each split and their averages with the standard deviations.

From the results, we notice similar trends as those deduced from separated $AP_{known}$ and $AP_{unk}$ evaluations. Yet, these trends become more distinct, offering a clearer understanding. Our baselines

Table 12: Detection accuracy for MTSD [3]. K, U1, and U2 stand for the splits of Known, Unknown1, and Unknown2, respectively.

| | K | U1 | U2 | U1+2 | mean |
|---|---|---|---|---|---|
| | $\mathrm{AP}_{known}$ | | $\mathrm{AP}_{unk}$ | | |
| ORE [9] | 41.2 | 0.4 | 0.2 | 0.7 | $0.4 \pm 0.3$ |
| DS [14] | 50.4 | 4.5 | 3.4 | 7.5 | $5.1 \pm 1.7$ |
| VOS [2] | 49.1 | 4.6 | 2.9 | 6.5 | $4.7 \pm 1.5$ |
| OpenDet [7] | 51.8 | 8.7 | 6.7 | 14.2 | $9.9 \pm 3.9$ |
| FCOS [15] | 41.7 | 3.8 | 3.3 | 6.2 | $4.4 \pm 1.6$ |
| Faster RCNN [14] | 50.0 | 2.5 | 2.3 | 4.4 | $3.1 \pm 1.2$ |

Table 13: H-scores for for Open Images dataset [10], "Animal" and "Vehicle" super-classes. See Table 10 for notations.

| | Animal | | | | |
|---|---|---|---|---|---|
| | Split1 | Split2 | Split3 | Split4 | mean |
| ORE [9] | 24.3 | 18.9 | 25.9 | 18.9 | $22.0 \pm 3.2$ |
| DS [12] | 26.5 | 18.4 | 21.2 | 21.4 | $21.9 \pm 2.9$ |
| VOS [2] | 24.3 | 20.3 | 21.9 | 24.5 | $22.7 \pm 1.7$ |
| OpenDet [7] | 38.3 | 24.4 | 40.3 | 36.0 | $34.8 \pm 6.2$ |
| FCOS [15] | 39.1 | 33.0 | 37.3 | 29.8 | $34.8 \pm 3.7$ |
| Faster RCNN [14] | 39.2 | 31.6 | 38.7 | 36.2 | $36.4 \pm 3.0$ |
| | Vehicle | | | | |
| | Split1 | Split2 | Split3 | Split4 | mean |
| ORE [9] | 1.0 | 0.2 | 0.4 | 0.6 | $0.5 \pm 0.3$ |
| DS [12] | 1.0 | 4.4 | 10.8 | 2.7 | $4.7 \pm 3.7$ |
| VOS [2] | 13.0 | 12.1 | 14.6 | 18.6 | $14.6 \pm 2.5$ |
| OpenDet [7] | 17.0 | 19.1 | 20.8 | 24.3 | $20.3 \pm 2.7$ |
| FCOS [15] | 22.1 | 20.2 | 21.6 | 21.3 | $21.3 \pm 0.7$ |
| Faster RCNN [14] | 17.4 | 22.3 | 25.8 | 25.8 | $22.8 \pm 3.4$ |

and OpenDet attain comparably better performances than other methods. Nonetheless, the resulting H-scores do not reach notably high values. This is attributed to the inferior performances of $\mathrm{AP}_{unk}$, largely deteriorate the harmonic mean of the known and unknown APs.

## D.3  Results of A-OSE and WI

In this study, we use the average precision for unknown object detection, denoted by $\mathrm{AP}_{unk}$, as a primary metric to evaluate OSOD methods, as reported in Table 3 in the main paper. For the readers' information, we report here absolute open-set error (A-OSE) and wilderness impact (WI), the metrics widely used in previous studies. Tables 16, 17, and 18 show those for the compared methods on the same test data. Recall that i) A-OSE and WI measure only detectors' performance of known object detection; and ii) they evaluate detectors' performance at a single operating point. Tables 16, 17, and 18 show the results at the operating points chosen in the previous studies, i.e., confidence score $> 0.05$ for A-OSE and the recall (of known object detection) $= 0.8$ for WI, respectively.

The results show that OpenDet and Faster RCNN achieve comparable performance on both metrics. FCOS performs worse, but this is not necessarily true at different operating points, as shown in Fig. 3 of the main paper. We can also see from the results a clear inconsistency between the A-OSE/WI and APs. For instance, as shown in Table 17, Faster RCNN is inferior to ORE in both the A-OSE and WI metrics (i.e., $6,382 \pm 206$ vs. $4,849 \pm 206$ on A-OSE), whereas it achieves much better $\mathrm{AP}_{known}$ and $\mathrm{AP}_{unk}$ than ORE, as shown in Table 11. Such inconsistency demonstrates that A-OSE and WI are unsuitable performance measures for OSOD-II/III.

Table 14: H-scores for CUB200 [16]. See Table 11 for notations.

|  | Split1 | Split2 | Split3 | Split4 | mean |
|---|---|---|---|---|---|
| ORE [9] | 26.8 | 31.0 | 26.7 | 30.8 | $28.8 \pm 2.1$ |
| DS [12] | 29.7 | 32.6 | 32.8 | 32.0 | $31.8 \pm 1.2$ |
| VOS [2] | 14.3 | 15.8 | 14.3 | 16.3 | $15.2 \pm 0.9$ |
| OpenDet [7] | 33.9 | 40.8 | 37.3 | 39.1 | $37.8 \pm 2.5$ |
| FCOS [15] | 32.4 | 35.4 | 33.5 | 33.6 | $33.7 \pm 1.1$ |
| Faster RCNN [14] | 32.0 | 37.0 | 34.8 | 35.2 | $34.8 \pm 1.8$ |

Table 15: H-scores for MTSD [3]. See Table 12 for notations.

|  | U1 | U2 | U1+2 | mean |
|---|---|---|---|---|
| ORE [9] | 0.8 | 0.4 | 1.4 | $0.9 \pm 0.4$ |
| DS [12] | 8.3 | 6.4 | 13.1 | $9.2 \pm 2.8$ |
| VOS [2] | 8.4 | 5.5 | 111.5 | $8.5 \pm 2.5$ |
| OpenDet [7] | 14.9 | 11.9 | 22.3 | $16.4 \pm 4.4$ |
| FCOS [15] | 7.0 | 6.1 | 10.8 | $8.0 \pm 2.0$ |
| Faster RCNN [14] | 4.8 | 4.4 | 8.1 | $5.7 \pm 1.7$ |

Table 16: A-OSE and WI of the compared methods in the experiment of Open Images. The same experimental setting as Table 10 is used.

| | Animal | | | | | | | | | |
|---|---|---|---|---|---|---|---|---|---|---|
| | Split1 | | Split2 | | Split3 | | Split4 | | mean | |
| | A-OSE | WI | A-OSE | WI | A-OSE | WI | A-OSE | WI | A-OSE | WI |
| ORE[9] | $23,334$ | 35.9 | $17,835$ | 30.8 | $22,219$ | 45.3 | $25,682$ | 47.0 | $22,268 \pm 2,848$ | $39.7 \pm 6.7$ |
| DS[12] | $44,377$ | 44.6 | $28,483$ | 38.6 | $39,592$ | 53.6 | $42,654$ | 63.6 | $38,776 \pm 6,185$ | $50.1 \pm 9.4$ |
| VOS [2] | $12,124$ | 34.8 | $21,622$ | 36.6 | $30,988$ | 50.9 | $23,360$ | 62.1 | $22,024 \pm 6,714$ | $46.1 \pm 11.2$ |
| OpenDet[7] | $26,426$ | 34.9 | $22,736$ | 27.7 | $25,075$ | 45.6 | $26,770$ | 56.1 | $25,252 \pm 1,585$ | $41.1 \pm 10.7$ |
| FCOS [15] | $38,858$ | 35.5 | $34,677$ | 37.6 | $52,234$ | 59.4 | $30,895$ | 49.5 | $39,166 \pm 8,053$ | $45.5 \pm 9.6$ |
| Faster RCNN [14] | $14,625$ | 30.9 | $11,121$ | 27.0 | $15,745$ | 46.8 | $16,260$ | 56.7 | $14,438 \pm 2,314$ | $40.4 \pm 13.8$ |
| | Vehicle | | | | | | | | | |
| | Split1 | | Split2 | | Split3 | | Split4 | | mean | |
| | A-OSE | WI | A-OSE | WI | A-OSE | WI | A-OSE | WI | A-OSE | WI |
| ORE [9] | $3,143$ | 17.6 | $3,775$ | 21.5 | $4,483$ | 33.7 | $6,654$ | 26.5 | $4,514 \pm 1,323$ | $24.9 \pm 6.0$ |
| DS [12] | $4,809$ | 22.7 | $10,617$ | 37.3 | $16,568$ | 53.6 | $12,107$ | 34.7 | $11,025 \pm 4,204$ | $37.1 \pm 11.0$ |
| VOS [2] | $1,460$ | 12.0 | $1,985$ | 23.9 | $1,796$ | 38.3 | $3,090$ | 20.9 | $2,083 \pm 611$ | $23.8 \pm 9.5$ |
| OpenDet [7] | $3,857$ | 19.8 | $5,640$ | 25.5 | $10,131$ | 52.1 | $8,893$ | 30.4 | $7,130 \pm 2,502$ | $31.9 \pm 12.2$ |
| FCOS [15] | $7,700$ | 26.4 | $10,888$ | 33.7 | $15,395$ | 55.7 | $22,502$ | 34.8 | $14,121 \pm 5,558$ | $37.6 \pm 10.9$ |
| Faster RCNN [14] | $3,487$ | 20.7 | $4,291$ | 25.4 | $6,138$ | 57.1 | $7,760$ | 31.7 | $5,444 \pm 1,956$ | $33.7 \pm 16.2$ |

Table 17: A-OSE and WI of the compared methods in the experiment of CUB200. The same experimental setting as Table 11 is used.

| | Split1 | | Split2 | | Split3 | | Split4 | | mean | |
|---|---|---|---|---|---|---|---|---|---|---|
| | A-OSE | WI | A-OSE | WI | A-OSE | WI | A-OSE | WI | A-OSE | WI |
| ORE[9] | $5,001$ | 22.6 | $4,836$ | 22.4 | $4,562$ | 24.1 | $4,998$ | 19.3 | $4,849 \pm 206$ | $22.1 \pm 2.0$ |
| DS[12] | $3,231$ | 17.4 | $3,567$ | 20.4 | $3,356$ | 21.8 | $3,301$ | 16.8 | $3,363 \pm 125$ | $19.1 \pm 2.1$ |
| VOS [2] | $4,681$ | 20.3 | $4,535$ | 21.0 | $4,763$ | 22.5 | $3,681$ | 18.5 | $4,415 \pm 498$ | $20.6 \pm 1.6$ |
| OpenDet[7] | $4,384$ | 18.6 | $4,746$ | 21.1 | $4,426$ | 22.6 | $4,602$ | 18.0 | $4,539 \pm 167$ | $20.1 \pm 2.2$ |
| FCOS [15] | $15,421$ | 24.1 | $18,334$ | 27.8 | $21,377$ | 25.8 | $16,822$ | 24.6 | $17,988 \pm 2,553$ | $25.6 \pm 1.6$ |
| Faster RCNN [14] | $5,898$ | 22.1 | $6,732$ | 24.0 | $6,289$ | 24.5 | $6,612$ | 20.0 | $6,382 \pm 206$ | $22.7 \pm 3.7$ |

Table 18: A-OSE and WI of the compared methods in the experiments of MTSD. The same setting is used as Table 12.

| | U1 | | U2 | | U1+2 | | mean | |
|---|---|---|---|---|---|---|---|---|
| | A-OSE | WI | A-OSE | WI | A-OSE | WI | A-OSE | WI |
| ORE[9] | $1,711$ | 5.5 | $2,050$ | 7.0 | $3,283$ | 11.7 | $2,348 \pm 827$ | $8.0 \pm 3.3$ |
| DS[12] | $1,658$ | 6.9 | $2,084$ | 8.4 | $3,742$ | 15.3 | $2,495 \pm 899$ | $10.2 \pm 3.7$ |
| VOS [2] | $1,260$ | 5.4 | $2,003$ | 8.9 | $3,263$ | 14.3 | $2,175 \pm 1,013$ | $9.5 \pm 4.5$ |
| OpenDet[7] | $722$ | 3.8 | $1,146$ | 7.8 | $1,868$ | 11.6 | $1,245 \pm 579$ | $7.8 \pm 3.9$ |
| FCOS[15] | $4,897$ | 5.7 | $7,086$ | 7.1 | $11,983$ | 12.8 | $7,989 \pm 3,628$ | $8.5 \pm 3.8$ |
| Faster RCNN[14] | $1,144$ | 5.5 | $1,702$ | 7.7 | $2,846$ | 13.2 | $1,897 \pm 868$ | $8.8 \pm 4.0$ |

Table 19: Results of the FCOS baseline with different values of $\gamma$ for each dataset. The numbers represent $\mathrm{AP}_{known}$ / $\mathrm{AP}_{unk}$ / WI. OI(A) and OI(V) indicate Open Images for Animal classes and Vehicle classes, respectively.

| Data \ $\gamma$ | 1.5 | 2.0 | 3.0 | 4.0 | 5.0 | 10.0 | 15.0 | 50.0 |
|---|---|---|---|---|---|---|---|---|
| OI(A) | 30.4 / 30.2 / 54.9 | 30.2 / 34.8 / 49.9 | 30.2 / 39.5 / 47.3 | 30.2 / 41.8 / 45.5 | 29.6 / 43.0 / 44.3 | 25.1 / 44.2 / 34.7 | 18.9 / 43.9 / 26.2 | 2.3 / 40.6 / 4.8 |
| OI(V) | 30.4 / 12.9 / 38.8 | 30.4 / 14.8 / 37.3 | 30.6 / 17.2 / 38.1 | 30.7 / 18.7 / 37.6 | 30.8 / 19.7 / 35.9 | 29.9 / 21.9 / 26.7 | 26.2 / 22.0 / 24.0 | 11.4 / 20.2 / 24.5 |
| CUB200 | 53.4 / 24.7 / 25.6 | 51.5 / 24.6 / 23.9 | 46.9 / 23.3 / 19.7 | 43.1 / 22.2 / 16.2 | 39.8 / 21.3 / 13.9 | 28.2 / 19.8 / 8.0 | 20.2 / 19.7 / 6.2 | 3.3 / 19.7 / 2.4 |
| MTSD | 41.7 / 4.4 / 8.5 | 39.5 / 5.2 / 9.5 | 36.7 / 6.0 / 10.4 | 34.3 / 6.3 / 8.5 | 32.3 / 6.5 / 7.6 | 25.4 / 6.4 / 4.1 | 21.6 / 6.2 / 3.4 | 8.6 / 5.5 / 0.7 |

Table 20: Results of the Faster RCNN baseline with different values of $\gamma$ and $T$ for each dataset. See Table 19 for notations.

| | Open Images (Animal) | | | | | | | |
|---|---|---|---|---|---|---|---|---|
| $T$ \ $\gamma$ | 1.5 | 2.0 | 3.0 | 4.0 | 5.0 | 10.0 | 15.0 | 50.0 |
| 0.5 | 32.5 / 8.4 / 60.6 | 32.5 / 11.8 / 60.5 | 32.5 / 15.0 / 60.2 | 32.4 / 16.6 / 60.0 | 32.4 / 17.6 / 59.9 | 32.2 / 20.1 / 59.4 | 32.1 / 21.0 / 59.3 | 31.6 / 23.6 / 59.1 |
| 0.8 | 39.5 / 17.6 / 54.6 | 39.5 / 21.4 / 54.2 | 39.4 / 24.3 / 53.6 | 39.2 / 25.8 / 53.0 | 39.0 / 26.8 / 52.6 | 38.4 / 29.4 / 50.8 | 37.9 / 30.8 / 49.0 | 37.0 / 34.0 / 46.4 |
| 1.0 | 40.6 / 22.7 / 51.1 | 40.5 / 25.9 / 50.5 | 40.0 / 28.5 / 49.3 | 39.6 / 30.1 / 47.9 | 39.3 / 31.3 / 46.6 | 38.4 / 33.9 / 42.6 | 37.8 / 35.3 / 40.4 | 36.2 / 37.6 / 35.7 |
| 2.0 | 38.5 / 22.4 / 27.9 | 36.9 / 24.5 / 27.3 | 34.5 / 26.2 / 25.1 | 31.8 / 26.6 / 24.1 | 28.4 / 26.3 / 20.9 | 9.8 / 23.4 / 11.1 | 2.3 / 22.7 / 4.1 | 0.0 / 22.7 / 0.0 |
| 3.0 | 20.0 / 15.5 / 14.9 | 14.3 / 16.8 / 17.9 | 3.6 / 16.1 / 9.0 | 0.1 / 15.7 / 0.0 | 0.0 / 15.7 / 0.0 | 0.0 / 15.7 / 0.0 | 0.0 / 15.7 / 0.0 | 0.0 / 15.7 / 0.0 |
| | Open Images (Vehicle) | | | | | | | |
| $T$ \ $\gamma$ | 1.5 | 2.0 | 3.0 | 4.0 | 5.0 | 10.0 | 15.0 | 50.0 |
| 0.5 | 33.2 / 2.1 / 46.2 | 33.2 / 3.2 / 46.0 | 33.2 / 4.1 / 46.0 | 33.2 / 4.8 / 46.1 | 33.2 / 5.1 / 45.9 | 33.2 / 6.1 / 45.9 | 33.2 / 6.7 / 45.9 | 33.2 / 7.9 / 45.6 |
| 0.8 | 39.8 / 5.9 / 40.7 | 39.8 / 7.6 / 40.4 | 39.8 / 9.1 / 40.2 | 39.8 / 10.0 / 40.0 | 39.8 / 10.6 / 39.7 | 39.7 / 12.2 / 39.1 | 39.6 / 13.1 / 38.8 | 39.5 / 15.2 / 37.8 |
| 1.0 | 40.4 / 9.0 / 37.2 | 40.4 / 10.8 / 36.9 | 40.3 / 12.4 / 36.8 | 40.3 / 13.4 / 36.3 | 40.3 / 14.1 / 36.1 | 40.1 / 16.1 / 34.4 | 39.9 / 17.0 / 33.7 | 39.4 / 19.4 / 30.8 |
| 2.0 | 40.3 / 15.5 / 27.0 | 40.0 / 17.9 / 25.3 | 39.1 / 20.3 / 21.5 | 37.3 / 21.1 / 19.7 | 34.9 / 21.1 / 13.7 | 13.6 / 18.5 / 12.0 | 1.8 / 17.9 / 7.6 | 0.0 / 17.9 / 0.0 |
| 3.0 | 13.0 / 17.1 / 6.7 | 7.2 / 19.5 / 13.7 | 3.8 / 17.9 / 9.2 | 0.0 / 17.5 / 0.0 | 0.0 / 17.5 / 0.0 | 0.0 / 17.5 / 0.0 | 0.0 / 17.5 / 0.0 | 0.0 / 17.5 / 0.0 |
| | CUB200 | | | | | | | |
| $T$ \ $\gamma$ | 1.5 | 2.0 | 3.0 | 4.0 | 5.0 | 10.0 | 15.0 | 50.0 |
| 0.5 | 58.0 / 14.0 / 25.3 | 57.9 / 17.6 / 25.2 | 57.9 / 20.7 / 25.2 | 57.8 / 21.7 / 25.0 | 57.7 / 22.3 / 24.9 | 57.3 / 23.2 / 24.9 | 57.1 / 23.8 / 24.8 | 56.6 / 24.1 / 24.4 |
| 0.8 | 61.9 / 20.7 / 23.2 | 61.8 / 22.9 / 23.2 | 61.7 / 23.8 / 23.1 | 61.6 / 24.2 / 22.9 | 61.5 / 24.3 / 22.7 | 60.7 / 24.1 / 21.8 | 59.7 / 23.8 / 20.6 | 57.7 / 23.1 / 17.9 |
| 1.0 | 62.3 / 22.8 / 22.9 | 62.3 / 23.8 / 23.0 | 62.2 / 24.2 / 22.7 | 62.0 / 23.9 / 22.4 | 61.7 / 23.8 / 22.2 | 60.1 / 23.2 / 20.3 | 58.3 / 22.6 / 18.6 | 53.7 / 20.9 / 13.9 |
| 2.0 | 61.8 / 20.9 / 18.9 | 59.6 / 19.9 / 17.5 | 53.0 / 18.2 / 12.6 | 46.1 / 17.6 / 8.7 | 39.5 / 17.5 / 6.2 | 15.6 / 17.4 / 2.2 | 4.7 / 17.4 / 1.5 | 0.0 / 17.4 / 0.0 |
| 3.0 | 10.7 / 0.3 / 2.5 | 10.3 / 0.2 / 2.4 | 5.1 / 0.2 / 1.2 | 0.4 / 0.2 / 2.1 | 0.0 / 0.2 / 0.0 | 0.0 / 0.2 / 0.0 | 0.0 / 0.2 / 0.0 | 0.0 / 0.2 / 0.0 |
| | MTSD | | | | | | | |
| $T$ \ $\gamma$ | 1.5 | 2.0 | 3.0 | 4.0 | 5.0 | 10.0 | 15.0 | 50.0 |
| 0.5 | 46.6 / 0.3 / 8.7 | 46.6 / 0.4 / 8.6 | 46.4 / 0.5 / 8.7 | 46.4 / 0.5 / 8.7 | 46.3 / 0.5 / 8.7 | 46.2 / 0.7 / 8.6 | 46.0 / 0.7 / 8.3 | 45.8 / 1.1 / 8.4 |
| 0.8 | 49.7 / 0.9 / 8.7 | 49.7 / 1.2 / 8.6 | 49.6 / 1.6 / 8.4 | 49.5 / 1.9 / 8.4 | 49.3 / 2.0 / 8.4 | 48.5 / 2.4 / 8.1 | 47.7 / 2.6 / 7.9 | 46.2 / 3.3 / 7.3 |
| 1.0 | 50.4 / 1.7 / 9.3 | 50.3 / 2.3 / 8.9 | 50.0 / 3.0 / 8.8 | 49.6 / 3.4 / 8.5 | 49.4 / 3.7 / 8.4 | 47.9 / 4.3 / 8.2 | 46.5 / 4.5 / 7.7 | 43.1 / 5.1 / 6.9 |
| 2.0 | 50.4 / 4.8 / 7.7 | 48.2 / 5.7 / 6.8 | 42.9 / 6.0 / 5.5 | 38.5 / 6.0 / 4.7 | 35.9 / 5.8 / 4.5 | 22.9 / 5.2 / 3.0 | 15.5 / 5.1 / 1.6 | 1.6 / 5.0 / 0.2 |
| 3.0 | 3.5 / 0.0 / 0.2 | 3.5 / 0.0 / 0.2 | 3.0 / 0.0 / 0.2 | 2.4 / 0.0 / 0.3 | 1.3 / 0.0 / 0.0 | 0.0 / 0.0 / 0.0 | 0.0 / 0.0 / 0.0 | 0.0 / 0.0 / 0.0 |

## D.4 Effect of Hyperparameters with the Baseline Methods

Our baseline methods use the ratio of the top two class scores for the known/unknown classification, where we use the hyperparameter $\gamma$ as a threshold. Tables 19 and 20 show how the choice of $\gamma$ affects the results. We can observe that overall, while $\gamma$ (and $T$ with Faster RCNN) do affect the results, $\mathrm{AP}_{known}$ and $\mathrm{AP}_{unk}$ are not very sensitive to their choice. There is a trade-off between $\mathrm{AP}_{known}$ and $\mathrm{AP}_{unk}$, since smaller $\gamma$ tends to make the detectors overlook unknown objects while large $\gamma$'s make the detectors overlook known objects. Setting a large temperature $T(> 1)$ with Faster RCNN damages performance on both $\mathrm{AP}_{known}$ and $\mathrm{AP}_{unk}$.

The optimal choice of the hyperparameters depends on datasets and model architectures. The dependency comes from two factors. One is the difference in the output layer design, i.e., sigmoid (FCOS) vs. softmax (Faster RCNN). Faster RCNN employs a softmax layer to predict the confidence scores, while FCOS uses a sigmoid layer. Due to the winner-take-all nature of softmax, Faster RCNN needs a relatively larger $\gamma$ to convert known predictions into unknown classes. The other is the number of classes in the datasets. Our configurations with CUB200 and MTSD have 150 and 230 of known classes, respectively, which are larger than that of Open Images (e.g., 24 classes for an "Animal" case). The larger the number of classes is, the more uncertain the prediction will be. Thus, small $\gamma$ is better for a small class set, and vice versa.

## D.5 Effects of Different Backbone Pretrained on a Large-Scale Data

Considering the recent success of open vocabulary detection (OVD) [17, 6, 13], we conjecture that utilizing a stronger backbone pre-trained on large-scale data could potentially enhance the performance of open-set object detection (OSOD). Thus, we conduct experimental evaluations using

109 such backbones on CUB200 and MTSD datasets, following the same experimental settings as above.
110 Specifically, we select OpenDet, which exhibits the best performance in our previous experiments.
111 OpenDet employs a ResNet50 model pre-trained on ImageNet-1K as its backbone. We replace it
112 with a ResNet50 model pre-trained on ImageNet-22K and fine-tune the entire detector (i.e., OpenDet)
113 on each dataset as usual.

114 Table 21 shows the results, which indicate that OpenDet with the new backbone produces better
115 $AP_{unk}$ on both datasets. This supports our conjecture, while the performance gain is modest. Futher
116 studies will be necessary.

Table 21: Effects of using different backbones on OSOD-III performance. OpenDet [7] adopting the standard backbone (ResNet50 pretrained on ImaegeNet1K) and a new backbone (ResNet50 pretrained an ImageNet22K) are compared. The average of all splits is reported.

| Training | IN22K | CUB200 [16] | | MTSD [3] | |
|---|---|---|---|---|---|
| | | $AP_{known}$ | $AP_{unk}$ | $AP_{known}$ | $AP_{unk}$ |
| OpenDet[7] | | $63.3 \pm 1.1$ | $27.0 \pm 3.0$ | $51.8$ | $9.9 \pm 3.9$ |
| | ✓ | $63.8 \pm 1.2$ | $27.3 \pm 3.4$ | $51.3$ | $11.0 \pm 4.4$ |

117 **D.6   More Examples of Detection Results**

118 Figures 6, 7, and 8 show more detection results for the four datasets, respectively. We only show the
119 bounding boxes with confidence scores $> 0.3$. We can observe from these results a similar tendency
120 to the quantitative comparisons we provide in the main paper. That is, OpenDet and our baselines
121 show comparable, limited performance in detecting unknown objects. They have the same several
122 types of erroneous predictions, such as failures to detect unknown objects, confusion of know objects
123 with unknown, and vice versa. Furthermore, they often predict two bounding boxes, significantly
124 overlapped, with known and unknown labels for the same object instances. Their limited performance
125 on $AP_{unk}$, along with these failures, indicates that the existing OSOD methods will be insufficient
126 for real-world applications.

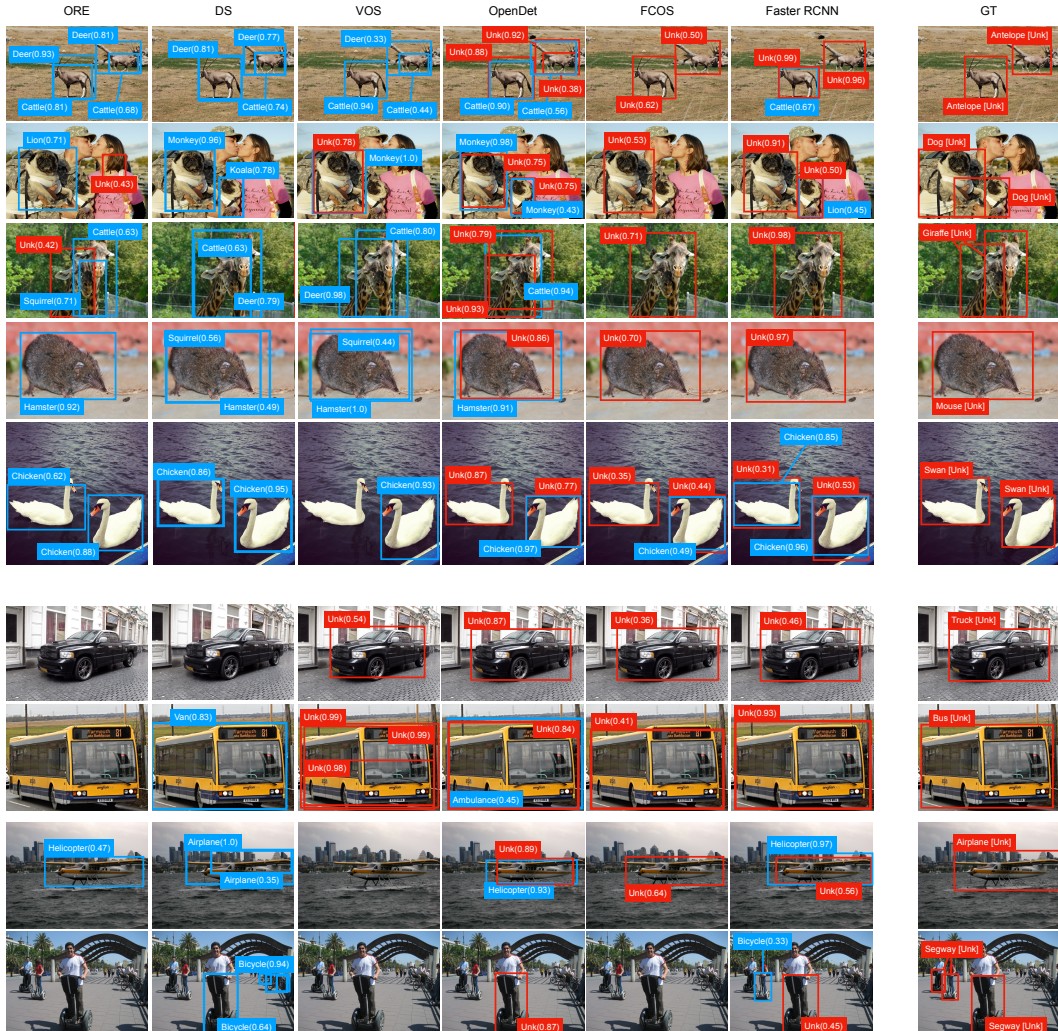

Figure 6: Examples of detection results for Open Images. Upper: the super-class is "Animal." Lower: "Vehicle." Red boxes represent unknown class detection, and blue boxes represent known class detection. "Unk" in the images stands for "unknown".

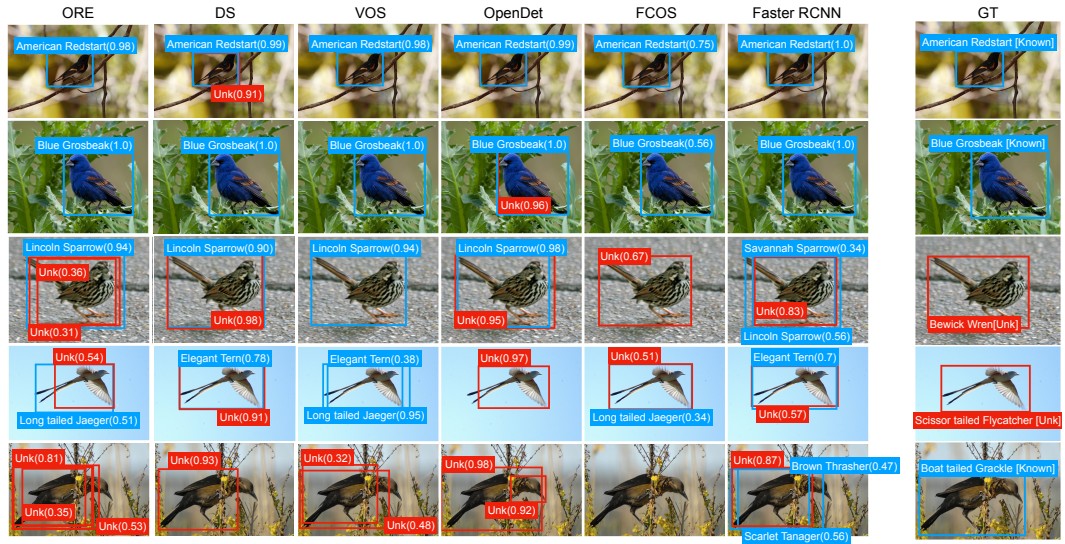

Figure 7: Examples of detection results for CUB200. See Fig 6 for notations.

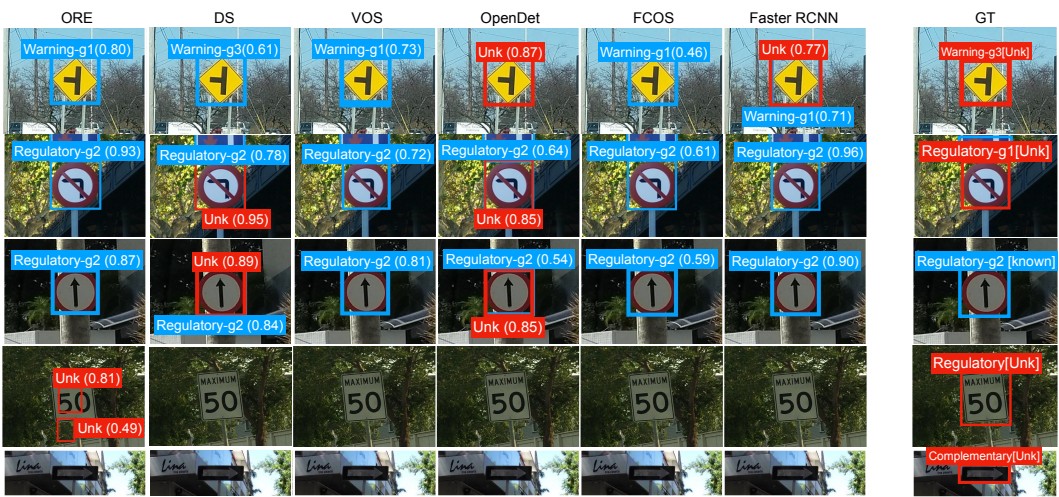

Figure 8: Examples of detection results for MTSD. See Fig 6 for notations.