# OpenReview forum: "Rectifying Open-Set Object Detection: Proper Evaluation and a Taxonomy"
_NeurIPS.cc/2023/Track/Datasets_and_Benchmarks — Submitted to NeurIPS 2023 Datasets and Benchmarks_

### Official Review · Reviewer_S4WT · 2023-07-10
**A good improvement in the definition and evaluation of Open Set Object Detection**

**Rating:** 7
**Confidence:** 4
**Correctness:** The claims are sound, the evaluation …
**Clarity:** The paper is well-written and well-st…

**Strengths:**

Main paper strengths:

 - the paper focuses on a problem that has a number of practical applications, but that till this point
   has not received enough attention, maybe precisely because a clear and well-defined, and
   constrained problem formulation and sane evaluation benchmarks were missing;
 - besides being easy to follow, the paper is well-written and well-structured: a taxonomy of
   current OSOD formulations is presented, their problems are analyzed, a new formulation is thus
   proposed and a novel benchmark is designed for it;
 - table 1 and figure 1 help a lot in clarifying the differences among the various problem
   formulations;
 - figure 2 and figure 3 clearly highlight the downsides of the OSOD-II formulation and the previous
   metrics used for OSOD;
 - the novel benchmark is well described and clearly allows to evaluate OSOD methods according to the novel OSOD-III problem
   formulation.

**Additional Feedback:**

The abstract could anticipate more explicitly which are the downsides of previous OSOD
formulations, for example, the fact that OSOD-II misses the definition of which unknown objects
should be detected.

**Documentation:**

The paper and supplementary material provide clear info on how to reproduce the experiments. A github repository with the needed code is also available

**Ethics:**

I do not see ethics concerns

**Limitations:**

The paper does not contain a limitation section and the limitations hence are not broadly discussed.

One of the possible limitations is however briefly mentioned by the authors at the end of section
4.2 and it is the fact that the whole proposed problem formulation is based on the concept of
super-class which is quite broad and not strictly mathematically defined. The proposed formulation
probably makes sense only as long as all the sub-classes that are part of a super-class are visually
similar. When this is not the case there is no reason why a detector should detect an unknown class.

In some applications, however, this condition may not hold. For example, in an autonomous driving
application, we may be interested in detecting and marking as unknown all those previously unseen objects that are on
the road and that may represent a safety concern, even when they do not share visual similarities with known classes.

Of course, even this one is a very specific scenario, thus my opinion is that even if the paper's definition
of which unknown objects should be detected may be useful only in some specific cases,
it is still an improvement over the previous situation of a completely missing definition of OSOD-II.


**Opportunities For Improvement:**

The main problems are in the evaluation metrics:

 - the authors argue that the metrics A-OSE and WI not only evaluate a single type of error but
   their measure is based on a single operating point, as they require to choose a confidence
   threshold a priori instead of using a threshold-free strategy as the Average Precision metric.
   This is certainly an interesting point but the authors' strategy to compute a confidence score
   for unknown samples is pretty arbitrary (line 242). Moreover, their evaluation strategy still
   requires to define a known-unknown threshold $\gamma$, whose definition may be quite tricky (which
   is evident from the fact that the authors use different thresholds for different datasets and
   base detectors, lines 244-247). This problem in OSR is solved by using the AUROC as
   a threshold-free metric to estimate the unknown detection ability of a model. A similar strategy
   could be employed here by requiring the detector to provide a normality score for each
   detection;
 - the tables report mAP for known classes and AP for the unknown one. While in most cases by looking
   at the numbers it is possible to certify when a model is better than another (i.e. when it
   outperforms the other in both metrics) in other situations this is not the case. It could be
   a good idea to also report a summary metric, for example, an harmonic mean of the two, similar
   to the H-score proposed in [A].

[A] Fu et al. "Learning to Detect Open Classes for Universal Domain Adaptation", ECCV 2020



**Relation To Prior Work:**

The paper proposes a taxonomy of the OSOD problem formulations clearly describing them and proposing an improvement over them.

**Summary And Contributions:**

This paper proposes a taxonomy of the current formulations of the Open Set Object Detection
problems, highlighting how the first proposed formulation (identified as OSOD-I) is pretty weak
as it focuses only on a single aspect of the problem (unknown objects detected as known), while the
second and most common formulation (OSOD-II) is fundamentally ill-posed, as it neglects the
differences between the Open Set Recognition (OSR) task and the OSOD one. OSR requires simply
marking all the objects as known or unknown, but it is a task designed on top of the classification
problem and thus considers single-object samples. This is of course not the
case for object detection, where detectors face multi-object images and are required to detect
only a subset of those objects. In OSOD-II instead, the detector is required to detect all objects,
both known and unknown ones, and no definition of what should be considered an "object" is provided.

The paper points out that this ill-posedness of previous OSOD formulations leads to the adoption of
performance metrics which are unsuited to represent and evaluate the OSOD performance of various
approaches. As a consequence, the paper proposes a novel OSOD formulation (OSOD-III) which is
well-constrained as it specifies exactly which kind of unknown objects should be detected, e.g. all
the ones that belong to the same same super-class of known classes.
Following this new formulation, the authors propose to discard the weak metrics employed that
previous OSOD studies and go back to the more standard object detection metric, the Average
Precision.

Once formalized the problem, the paper proposes a novel benchmark designed on top of this
formalization and based on 3 publicly available datasets. A picture of the state-of-the-art on this
novel benchmark is drawn by evaluating on it the performance of a bunch of OSOD-II methods and
a newly proposed benchmark. The experimental analysis highlights that the current OSOD approaches
are still unable to obtain good results on this difficult task and are even often unable to
outperform a simple baseline.

Contributions:

 - a taxonomy of OSOD formulations
 - a novel formulation overcoming the problems of the previous ones;
 - a novel benchmark designed on top of the novel problem formulation;
 - a picture of the state-of-the-art.

---

> ### Author Response · Authors · 2023-08-18
> **Author response to Reviewer S4WT**
>
> **1. The authors argue that the metrics A-OSE and WI not only evaluate a single type of error … A similar strategy could be employed here by requiring the detector to provide a normality score for each detection;**
>
> Indeed, the method we introduce as a "naive baseline" uses an arbitrary computation for the confidence score of unknown instances. The method requires specifying a threshold to classify a prediction as either known or unknown. However, we emphasize that this threshold is not used for evaluating any detector, including the baseline method. It is employed within the baseline method to distinguish between known and unknown instances. Thus, our main metric, AP, is independent of any manually specified threshold.
> Furthermore, we recognize that certain methods for OSR might be adapted for OSOD to enhance detection performance and to devise superior evaluation metrics. We consider this a direction for future research.
>
> **2. The tables report mAP for known classes and AP for the unknown one. … It could be a good idea to also report a summary metric, for example, an harmonic mean of the two, similar to the H-score proposed in [A].**
>
> Thank you for the suggestion. We have calculated the H-scores for the detectors tested; please refer to the table below. This result provides additional clarity to our arguments. We will include this table in the supplementary material.
>
> |             | OI-Animal     | OI-Vehicle    | CUB200        | MTSD          |
> | ----------- | ------------- | ------------- | ------------- | ------------- |
> | ORE         | $22.0\pm 3.2$ | $0.5\pm 0.3$  | $28.2\pm 2.1$ | $0.9\pm 0.4$  |
> | VOS         | $22.7\pm 1.7$ | $14.6\pm 2.5$ | $15.2\pm 2.1$ | $8.5\pm 2.5$  |
> | OpenDet     | $34.8\pm 6.2$ | $20.3\pm 2.7$ | $37.8\pm 2.5$ | $16.4\pm 4.4$ |
> | FCOS        | $34.8\pm 3.7$ | $21.3\pm 0.7$ | $33.7\pm 1.1$ | $8.0\pm 2.0$  |
> | Faster RCNN | $36.4\pm 3.0$ | $22.8\pm 3.4$ | $34.8\pm 1.8$ | $5.7\pm 1.7$  |

---

> ### Comment · Reviewer_S4WT · 2023-08-25
> **Final rate**
>
> After reading the other reviews and the authors' responses I have decided to keep my original rating of 7

---

### Official Review · Reviewer_kthN · 2023-07-20
**A reasonable new problem setting for open-set object detection, but lacking analysis and insights on experimental results.**

**Rating:** 6
**Confidence:** 4
**Clarity:** Yes. It is clear and well written.

**Strengths:**

++ The problem setting addresses the issue of the current OSOD setting (mainly OSOD-II) that it is unclear and ambiguous about what should be detected as an “unknown” object. I think having an unambiguous problem setting for a research field is important, otherwise the goal of this field would be vague, and the experimental evaluations conducted in such an unclear setting would be not convincing and unreliable. The authors take a meaningful step to set up a new problem setting.

++ The findings on the experimental evaluation in this new problem setting is interesting and surprising. It shows that the current OSOD methods have worse performance than some naïve and simple baselines that only use conventional detectors (FCOS, Faster RCNN) in detecting unknown. It shows that further investigation is needed in the proposed new problem setting.

++ The presentation of this paper is good. I feel that I could easily understand the motivation of proposing the new OSOD problem setting, how the new problem setting is different from previous settings, and the advantage of the proposed setting over the previous settings. Also, the taxonomy of different OSOD problem settings made in the paper is clear.


**Additional Feedback:**

I would like to see the authors’ analysis and insight about the findings of experimental results in the new proposed settings. Also, authors are encouraged to address the concern regarding the NMS post processing.

**Correctness:**

Yes. The details of the datasets and experiment designs are contained in the supplementary.

**Documentation:**

Yes. The authors provide a valid URL for dataset and include details for experiments as well.

**Ethics:**

No. The datasets are constructed on existing datasets, and the existing datasets also have low risk in ethical issues.

**Limitations:**

Yes. The authors have discussed the limitations of the unsatisfactory results of current OSOD methods in the new problem setting (Sec. 4.3). There is no evident potential negative societal impact of this work.

**Opportunities For Improvement:**

-- My biggest concern of this paper is that there is little analysis and insight on the experimental results that could benefit future research. The authors show that current SOTA OSOD methods fail in the new OSOD-III setting, but there is no analysis on “why they fail”, “why naïve baselines could even have better performance”. It is unclear that whether the reason is (a) the current SOTA methods are specially designed for OSOD-II setting, so they do not adapt well in OSOD-III setting; or (b) the OSOD-III setting also has some improper aspects; or there are other reasons. Also, the authors did not provide much insight in “how we could design algorithms in the new OSOD-III setting”, “what would be the potential key and direction in successfully handling the OSOD-III problem”. I understand that this is a new setting, so limited preliminary experiments are conducted, but such analysis and insight would be helpful for researchers to do follow-up works in the proposed new setting. Otherwise, the impact and contribution of this paper would be limited.

-- From the qualitative results in the paper (e.g., Fig. 4) and supplementary material (e.g., Fig. 5), it seems that the results have not been post-processed by non-maximum suppression (NMS). If NMS is applied, I think the performance for many methods would be improved. For example, in Fig. 4 in the main paper, for the first image of OpenDet, the bounding box which has the label “Cattle (0.89)” would be filtered because it has large IOU with “Unk (0.93)” but has lower confidence. Did the authors use NMS for post-processing? If not, I think the authors could use NMS on all the methods to see whether the conclusion made in this paper would be changed.

-- Overall, the presentation of this paper is very good. But one minor suggestion may be that in the abstract, I find myself kind of lost because the authors did not mention what the “fundamental issue” is throughout the whole abstract, so it is a little confusing. Only until the introduction part did the authors mention that the issue is the ambiguity of the definition of “objects”. I think it might be better to include “what the fundamental issue” is directly in the abstract.


**Relation To Prior Work:**

Yes. The authors have clearly discussed how the proposed problem setting is different from previous problem settings, and how methods designed for previous problem settings fail in the proposed problem setting.

**Summary And Contributions:**

This paper proposes a new open-set object detection (OSOD) problem setting, which differs from two kinds of mainstream OSOD problem settings: (1) detecting all instances of known objects without being distracted by unknown objects (OSOD-I); (2) detecting all instances of both known and unknown objects, classifying unknown objects into the “unknown” category (OSOD-II). Mainly due to the vague and ambiguous definition of what should be an “object”, the authors propose a new problem setting which treats only the unknown subclasses that belong to a known super-class as the “unknown” objects that need to be detected. The authors evaluate the existing OSOD methods on the new problem setting but find that they surprisingly have worse performance than some naïve baselines.

---

> ### Author Response · Authors · 2023-08-18
> **Author response to Reviewer kthN**
>
> **1. My biggest concern of this paper is that there is little analysis and ... Otherwise, the impact and contribution of this paper would be limited.**
>
> In response to the comment, we have conducted an additional analysis. Please refer to our post “Further analysis on failures to detect unknown instances” at the top of this page.
> In summary, the primary reason existing SOTA OSOD methods fall short for OSOD-III is their frequent inability to accurately classify known and unknown instances. This discrepancy arises because their visual characteristics are more similar in OSOD-III than in the standard experimental settings of OSOD-II. Hence, the most fitting answer from the provided options would be (a).
> It is important to note that simply applying NMS to known and unknown predictions does not address the problem. This is likely due to challenges in comparing scores between known and unknown predictions, as they may not be aptly calibrated for direct comparison.
> These insights emphasize the challenges introduced by OSOD-III and indicate a promising direction for subsequent research.
> We have integrated (or will soon integrate) this analysis into our manuscript.
>
>
> **2. From the qualitative results in the paper (e.g., Fig. 4) ... whether the conclusion made in this paper would be changed.**
>
> As previously noted, applying NMS to both known and unknown predictions does not enhance detection accuracy. In fact, it reduces AP for both instance types.
> To clarify, in the results presented in our initial manuscript, we applied NMS to predictions within distinct categories, following the standard procedure in object detection. Specifically, NMS was executed for each known category, as well as for the unknown category. However, the appropriateness of treating the unknown "category" on par with known categories can be debated. Consequently, we assessed NMS across both known and unknown predictions, obtaining the above finding. Please see our post “Further analysis on failures to detect unknown instances” for more details.
>
>
> **3. Overall, the presentation of this paper is very good. ... I think it might be better to include “what the fundamental issue” is directly in the abstract.**
>
> Thank you for your feedback. We will update the abstract to resolve the concern.

---

> > ### Comment · Reviewer_kthN · 2023-08-30
> > **Review update after rebuttal**
> >
> > Thanks the authors for giving further analysis on the results and providing additional experiments about NMS, which makes the benchmark on the new OSOD setting more solid and convincing. I keep my original score.

---

### Official Review · Reviewer_wjZb · 2023-07-20
**Nice benchmark for OSOD.**

**Rating:** 7
**Confidence:** 3
**Correctness:** The evaluation protocols and experime…
**Clarity:** The paper is well written and easy to…

**Strengths:**

- The proposed evaluation protocol for OSOD is more convincing and feasible than previous evaluation protocols.

- The authors describe the limitations of previous evaluation protocols in detail. With this paper, even the newcomers in this field can understand the OSOD research field very well.

- With the proposed evaluation protocol, the authors found that existing methods are not effective for solving OSOD problems actually. This will shed light on a new research direction of OSOD research field.

**Additional Feedback:**

N/A

**Documentation:**

There are sufficient details to support reproducibility.

**Limitations:**

I believe that, in autonomous driving, it might be dangerous if the detector only cares about the classes within specific superclasses. In such situation, it is worthy of detecting any kind of objects.

**Opportunities For Improvement:**

- The authors only utilize four existing methods. It would be great if the authors evaluate more existing methods with the evaluatoin metrics.

- It would be great if the authors analyze why the existing methods are not good at OSOD in both quantitative and qualitative ways.



**Relation To Prior Work:**

There are good discussion about the relationship to previous evaluation protocols.

**Summary And Contributions:**

The authors propose a novel evaluation protocol for Open-Set Object Detection (OSOD). To address the infeasibility of previous evaluation metrics for OSOD, the proposed protocol is only handling the unknown classes within specified super-classes of known object classes. With this, the ill-posedness of conventional evaluation metrics can be resolved. They also propose new evaluation metrics to measure the performances of OSOD methods more faithfully. With the evaluation protocol, the authors re-evaluate previous methods and observe that they are inferior to a simple baseline which is proposed by the authors.

---

> ### Author Response · Authors · 2023-08-18
> **Author response to Reviewer wjZb**
>
> **1. The authors only utilize four existing methods. It would be great if the authors evaluate more existing methods with the evaluatoin metrics.**
>
> In response to your comment, we add the results of Dropout Sampling (DS) [A] for OSOD-III tasks; see the table below. Note that our claims remain the same.
>
> |                     | OI-Animal          |                  | OI-Vehicle         |                  | CUB200             |                  | MTSD               |                  |
> | ------------------- | ------------------ | ---------------- | ------------------ | ---------------- | ------------------ | ---------------- | ------------------ | ---------------- |
> |  | ${\rm AP}_{known}$ | ${\rm AP}_{unk}$ | ${\rm AP}_{known}$ | ${\rm AP}_{unk}$ | ${\rm AP}_{known}$ | ${\rm AP}_{unk}$ | ${\rm AP}_{known}$ | ${\rm AP}_{unk}$ |
> | ORE | $37.6\pm 2.8$      | $15.6\pm 2.7$    | $33.7\pm 8.5$      | $0.3\pm 0.1$     | $53.2\pm 1.3$      | $19.8\pm 2.2$    | $41.2$             | $0.4\pm 0.3$     |
> | **DropoutSampling[A]** | **41.1$\pm$ 2.9**      | **15.0$\pm$ 2.5**    | **40.1$\pm$ 7.9**      | **2.7$\pm$ 2.3**     | **61.5$\pm$ 0.9**      | **21.5$\pm$ 1.1**    | **50.4**             | **5.1$\pm$ 1.7**     |
> | VOS | $39.5\pm 2.2$      | $16.0\pm 1.8$    | $40.9\pm 7.8$      | $9.1\pm 2.2$     | $59.4\pm 1.0$      | $8.7\pm 0.6$    | $49.1$             | $4.7\pm 1.5$     |
> | OpenDet | $36.9\pm 8.1$      | $33.0\pm 4.5$    | $38.7\pm 7.8$      | $14.4\pm 3.3$     | $63.3\pm 1.1$      | $27.0\pm 3.0$    | $51.8$             | $9.9\pm 3.9$     |
> | FCOS | $30.3\pm 4.7$      | $41.8\pm 3.6$    | $30.7\pm 12.0$      | $18.7\pm 4.5$     | $53.5\pm 2.1$      | $24.7\pm 1.3$    | $41.7$             | $4.4\pm 1.6$     |
> | Faster RCNN | $37.8\pm 3.1$      | $35.3\pm 3.9$    | $39.9\pm 8.7$      | $17.0\pm 5.2$     | $62.2\pm 1.0$      | $24.2\pm 1.9$    | $50.0$             | $3.1\pm 1.2$     |
>
> [A] D. Miller et al., “Dropout Sampling for Robust Object Detection in Open-set Conditions”, In Proc. ICRA, 2018.
>
>
> **2. It would be great if the authors analyze why the existing methods are not good at OSOD in both quantitative and qualitative ways.**
>
> Please see our post “Further analysis on failures to detect unknown instances” at the top of this page. It delves into the reasons why existing detectors often struggle to identify unknown instances. We have incorporated (or will incorporate) the analysis into our manuscript.
>
> **3. I believe that, in autonomous driving, it might be dangerous if the detector only cares about the classes within specific superclasses. In such situation, it is worthy of detecting any kind of objects.**
>
> We concur. Detecting every "unusual" object, whether stationary or moving on the road, might be crucial for autonomous driving. Yet, OSOD-III does not address this particular scenario. However, OSOD-III can effectively deal with the problem of properly handling known and unknown traffic signs. This is an existing concern in the industry and has been overlooked in the research community.

---

### Official Review · Reviewer_ariW · 2023-07-21
**The paper proposed a new setting for Open-Set Object Detection due to the limitation of existing settings.**

**Rating:** 5
**Confidence:** 4
**Correctness:** The claims are correct
**Clarity:** The paper is clear, though the logic …

**Strengths:**

1. The proposed setting is better defined than the existing open-set object detection setting.


**Additional Feedback:**

No

**Documentation:**

The benchmark is based on existing released datasets by modifying the labels for new settings.

**Opportunities For Improvement:**

1. The significance of the new setting for OSOD. Recent detection methods mainly focus on open-vocabulary object detection, which has broader applications and is more challenging than open-set detection. Furthermore, the proposed setting, e.g., detect animal (super class of both known and unknown class) after trained on detecting dogs, shares a similar spirit of open-vocabulary object detection, which aims to detect objects of classes described by texts in inference-time.
2. Given the existence of open-vocabulary object detection, the paper might need to highlight the significance of improvement over OSOD so as its future research directions.
3. No further insights is provided after benchmarking methods on the new setting. The paper should point out the challenge of the new setting so as the future research direction on this field.

**Relation To Prior Work:**

Yes.

**Summary And Contributions:**

This paper analyzes the limitation of existing open-set object detection benchmarks, metrics, and settings, and proposes a new benchmark termed as OSOD-III, which aims to detect all objects from the super-classes of a closed set known class.

---

> ### Author Response · Authors · 2023-08-18
> **Author response to Reviewer ariW**
>
> **1. The significance of the new setting for OSOD. Recent detection methods mainly focus on open-vocabulary object detection, which has broader applications and is more challenging than open-set detection. Furthermore, the proposed setting, e.g., detect animal (super class of both known and unknown class) after trained on detecting dogs, shares a similar spirit of open-vocabulary object detection, which aims to detect objects of classes described by texts in inference-time. Given the existence of open-vocabulary object detection, the paper might need to highlight the significance of improvement over OSOD so as its future research directions.**
>
> While it's true that an ideal OVD could effectively handle unknown categories, it is a misconception to assume that a real OVD method would entirely replace the need for OSOD. Here's why:
> - Addressing both known and unknown categories across diverse applications often requires vast amounts of data. Many current studies primarily utilize data from object categories that are readily available on the Internet, which may not capture the full breadth of real-world objects.
> - Even if we overcome the data challenge, the resulting detectors would likely have significantly large model sizes. Such oversized models are impractical for many real-world scenarios. For instance, a traffic sign detector in a car's driver assistance system must remain compact.
> - Related to the first point, OVD's capability is limited to categories present in the training data. Certain applications need to adapt to continuously evolving categories, such as manufactured goods. For instance, new traffic signs might emerge monthly worldwide.
>
> Given these challenges, irrespective of the advances in OVD research, there will always be a demand for OSOD. One might rather argue that OVD's applicability in addressing real-world problems is limited.
>
>
> **2. No further insights is provided after benchmarking methods on the new setting. The paper should point out the challenge of the new setting so as the future research direction on this field.**
>
> Please see our post “Further analysis on failures to detect unknown instances” at the top of this page. It delves into the reasons why detectors often struggle to identify unknown instances. This analysis sheds light on the challenges posed by OSOD-III and suggests a direction for future research. We have incorporated (or will incorporate) these insights into our manuscript.

---

### Official Review · Reviewer_kbs8 · 2023-07-25
**NeurIPS 2023 Track Datasets and Benchmarks Submission443 Reviewer kbs8**

**Rating:** 5
**Confidence:** 3
**Clarity:** The paper is well written.

**Strengths:**

1. Restricting the identification of unknown classes to superclasses of known classes is a reasonable and relevant setup.
2. A new way of constructing benchmark is proposed


**Additional Feedback:**

1. Is there a reason why the author's newly constructed dataset doesn't use the common detection datasets, coco and voc?
2. WI and AOSE were not used in the assessment and were not comprehensive enough, or there may be more accurate measures of incorrect scores.


**Correctness:**

The datasets are constructed in sound way. But the WI and AOSE were not used in the assessment and were not comprehensive enough.

**Documentation:**

The benchmarks have sufficient detail to support reproducibility.

**Ethics:**

There aren't any ethical concerns with the submission.

**Limitations:**

1. WI and AOSE were not used in the assessment and were not comprehensive enough, or there may be more accurate measures of incorrect scores.

2. Even with the second setup, does it matter if only labelled categories are counted in the test, the authors need to give more detail on this.
3. Is there a reason why the authors' newly constructed dataset doesn't use the common detection datasets, coco and voc?
4. More models should be added in the experiments.


**Opportunities For Improvement:**

1. WI and AOSE were not used in the assessment and were not comprehensive enough, or there may be more accurate measures of incorrect scores.
2. Even with the second setup, does it matter if only labelled categories are counted in the test, the authors need to give more detail on this.
3. Is there a reason why the authors' newly constructed dataset doesn't use the common detection datasets, coco and voc?


**Relation To Prior Work:**

This work clearly discusses the difference with previous contributions

**Summary And Contributions:**

This paper proposes a novel formulation where detectors are required to detect both known and unknown classes within specified super-classes of object classes. And the authors design benchmark tests utilizing existing datasets and report the experimental evaluation of existing OSOD methods.

---

> ### Author Response · Authors · 2023-08-18
> **Author response to Reviewer kbs8**
>
> **1. WI and AOSE were not used in the assessment and were not comprehensive enough, or there may be more accurate measures of incorrect scores.**
>
> In the supplementary material, we've included the scores of WI and AOSE; see Tables 13, 14, and 15 in Sec. D.2. However, it's important to highlight our main claim: WI and AOSE are not valid measures for OSOD-II/III. These scores are provided solely for comparison with previous studies.
>
> **2. Even with the second setup, does it matter if only labelled categories are counted in the test, the authors need to give more detail on this.**
>
> We can compute the mAP you suggested (if we interpret your comment correctly). However, this mAP score would be for a predefined set of unknown categories, suggesting that we were aware of which categories were unknown beforehand. This contradicts the very definition of open-setness. This is one of our main claims. Please refer to Sec. 2.1.
>
> **3. Is there a reason why the authors' newly constructed dataset doesn't use the common detection datasets, COCO and VOC?**
>
> In OSOD-III, a dataset must have a hierarchical structure where both known and unknown categories are subclasses of the same superclass. While OpenImage, CUB200, and MTSD possess these rich hierarchies, COCO and PASCAL-VOC are lacking in this regard.
>
> **4. More models should be added in the experiments.**
>
> In response to your comment, we add the results of Dropout Sampling (DS) [A] for OSOD-III tasks; see the table below. Note that our claims remain the same.
>
> |                     | OI-Animal          |                  | OI-Vehicle         |                  | CUB200             |                  | MTSD               |                  |
> | ------------------- | ------------------ | ---------------- | ------------------ | ---------------- | ------------------ | ---------------- | ------------------ | ---------------- |
> |  | ${\rm AP}_{known}$ | ${\rm AP}_{unk}$ | ${\rm AP}_{known}$ | ${\rm AP}_{unk}$ | ${\rm AP}_{known}$ | ${\rm AP}_{unk}$ | ${\rm AP}_{known}$ | ${\rm AP}_{unk}$ |
> | ORE | $37.6\pm 2.8$      | $15.6\pm 2.7$    | $33.7\pm 8.5$      | $0.3\pm 0.1$     | $53.2\pm 1.3$      | $19.8\pm 2.2$    | $41.2$             | $0.4\pm 0.3$     |
> | **DropoutSampling[A]** | **41.1$\pm$ 2.9**      | **15.0$\pm$ 2.5**    | **40.1$\pm$ 7.9**      | **2.7$\pm$ 2.3**     | **61.5$\pm$ 0.9**      | **21.5$\pm$ 1.1**    | **50.4**             | **5.1$\pm$ 1.7**     |
> | VOS | $39.5\pm 2.2$      | $16.0\pm 1.8$    | $40.9\pm 7.8$      | $9.1\pm 2.2$     | $59.4\pm 1.0$      | $8.7\pm 0.6$    | $49.1$             | $4.7\pm 1.5$     |
> | OpenDet | $36.9\pm 8.1$      | $33.0\pm 4.5$    | $38.7\pm 7.8$      | $14.4\pm 3.3$     | $63.3\pm 1.1$      | $27.0\pm 3.0$    | $51.8$             | $9.9\pm 3.9$     |
> | FCOS | $30.3\pm 4.7$      | $41.8\pm 3.6$    | $30.7\pm 12.0$      | $18.7\pm 4.5$     | $53.5\pm 2.1$      | $24.7\pm 1.3$    | $41.7$             | $4.4\pm 1.6$     |
> | Faster RCNN | $37.8\pm 3.1$      | $35.3\pm 3.9$    | $39.9\pm 8.7$      | $17.0\pm 5.2$     | $62.2\pm 1.0$      | $24.2\pm 1.9$    | $50.0$             | $3.1\pm 1.2$     |
>
> [A] D. Miller et al., “Dropout Sampling for Robust Object Detection in Open-set Conditions”, In Proc. ICRA, 2018.

---

### Author Response · Authors · 2023-08-18
**Further analysis on failures to detect unknown instances**

In response to the reviewers’ comment, we provide further analysis on the failure modes of the tested detectors for the OSOD-III tasks. Our findings indicate that while these detectors demonstrate a reasonably high recall for detecting unknown instances, they often confuse known and unknown instances. Specifically, they frequently misclassify known instances as unknown.

The details are as follows. In our manuscript's experiments, we individually applied NMS to each category, both known and unknown. This aligns with the standard object detection practice where NMS is primarily used among the predicted bounding boxes (BBs) of a specific category. However, the appropriateness of treating the unknown category in the same way as known categories can be debated. Therefore, we also conducted NMS across both known and unknown category predictions to assess its impact on detection accuracy. Tables A and B show the mAP for the known category predictions and AP for the unknown, evaluated across different IoU thresholds for NMS. Here, NMS is performed on predicted BBs based on the higher IoU values between known and unknown categories.

In Tables A and B, an IoU threshold of 1.0 represents results obtained without NMS, while other values depict results with NMS application. It's evident that a more aggressive NMS leads to lower APs for both the known and unknown categories. This reveals two points: i)​​ The predicted known and unknown BBs have overlaps frequently, and ii) The scores of these bounding boxes don't align well with the accuracy of the predictions. Ideally, among overlapping BBs, the one with the highest score should represent the correct prediction. However, in practice, bounding boxes that misidentify known instances as unknown often have higher scores than those that correctly identify them as known. Conversely, boxes that mistake unknown instances for known ones sometimes score higher than their accurate counterparts.

Future research should address the prevalent issue of misclassifying known and unknown instances. While detecting BBs of unknown instances isn't particularly challenging, the issue arises in classification: BBs predicted for unknown instances are often mislabeled as known, and vice-versa. Moreover, simply applying NMS to both known and unknown predictions isn't a comprehensive solution. The primary challenge appears to be in comparing their respective confidence scores. This discrepancy is likely because the scores aren't consistently calibrated between the known and unknown categories.

Table A: Detection accuracy on CUB200 across various IoU thresholds for NMS.
|   | NMS IoU threshold | 0.5               | 0.7  | 0.8  | 0.9  | 1.0 (w/o NMS) |
| ----------- | ------------------ | ---------------- | --- | ---- | ---- | ------------- |
| ORE         | ${\rm AP}_{known}$ | 51.7              | 51.8 | 51.9 | 52.6 | 53.2          |
|             | ${\rm AP}_{unk}$   | 11.9              | 12.1 | 13.1 | 17.2 | 19.8          |
| OpenDet     | ${\rm AP}_{known}$ | 48.2              | 48.8 | 50.3 | 54.4 | 63.3          |
|             | ${\rm AP}_{unk}$   | 25.0              | 24.9 | 24.9 | 25.3 | 27.0          |
| Faster RCNN | ${\rm AP}_{known}$ | 50.5              | 50.5 | 50.7 | 51.7 | 62.2          |
|             | ${\rm AP}_{unk}$   | 17.2              | 17.2 | 17.3 | 17.9 | 24.2          |

Table B: Detection accuracy on MTSD across various IoU thresholds for NMS.
|   | NMS IoU threshold | 0.5  | 0.7  | 0.8  | 0.9  | 1.0 (w/o NMS) |
| ----------- | ------------------ | --- | --- | ---- | ---- | ------------- |
| ORE         | ${\rm AP}_{known}$ | 40.6 | 40.8 | 40.9 | 41.1 | 41.2          |
|             | ${\rm AP}_{unk}$   | 0.3  | 0.3  | 0.3  | 0.4  | 0.4           |
| OpenDet     | ${\rm AP}_{known}$ | 34.4 | 34.6 | 35.0 | 37.9 | 51.8          |
|             | ${\rm AP}_{unk}$   | 10.0 | 10.0 | 9.9  | 9.6  | 9.9           |
| Faster RCNN | ${\rm AP}_{known}$ | 44.1 | 44.1 | 44.1 | 44.6 | 50.0          |
|             | ${\rm AP}_{unk}$   | 0.9  | 0.9  | 0.9  | 1.0  | 3.1           |

---

### Decision · Program_Chairs · 2023-09-22

**Decision:**

Reject

**Comment:**

This paper receives mixed reviews, and is on the borderline for acceptance. Although the AC believes the recognized flaw in existing OSOD pipeline is a novel contribution (S4WT, wjZb, kthN), the underlying impact with the "corrected" version of evaluation pipeline is not very well justified (kthN, kbs8, ariW, S4WT). As a result, the work does not meet the bar for acceptance. AC recommends the authors to review the suggestions from reviewers and consider re-submitting the work in the future.

Extended Review:

This paper proposes a new evaluation protocol for OSOD, which recognizes the underlying ambiguity in the taxonomy of object to detect in the previous OSOD evaluation. The author proposed to constraint the evaluation of OSOD to a "super-class". Although there is a decent amount of contributions from the new evaluation protocol, the work mainly suffers from a lack of analysis on the practical impact with the newly proposed benchmarking system. For example, the super-class is somewhat undefined (S4WT). AC believes that some empirical analysis on what makes a good super-class would be useful (i.e. it may fall within a spectrum between a single category to all objects (OSOD-II setup)). More insights regarding this perspective may help alleviate concerns in adoption to other dataset (ariW, e.g. how to generate super-class in COCO/ LVIS, or why this is not feasible). In addition, a concern regarding the practical value compared to Open-vocab object detection (ariW) is also worth more investigation. For example, is there a way to incorporate the proposed change to OVOD evaluation? Or can the analysis drawn from the proposed evaluation benefit other tasks?